**communications** sustainability

# Human contributions to evapotranspiration mitigate swings in dry-to-wet year transitions
Zoe Amie Pierrat [1] ✉, Rebecca N. Gustine [1,2], Anna Boser[3], Sophie Ruehr[4], Christine M. Lee [1] ✉,
J. T. Reager [1] & Kerry Cawse-Nicholson [1]

California's food and economic security depends on water availability, particularly under increasingly extreme climate scenarios. A key component of the water balance is evapotranspiration, the combination of soil and surface evaporation and plant transpiration. Evapotranspiration is influenced by natural drivers (e.g., climate, vegetation cover) and human intervention (e.g., irrigation, land management). Here, we analyze the transition between one of California's driest years (2022) to an exceptionally wet year (2023) to assess evapotranspiration responses to climate extremes. Despite increased precipitation, total statewide evapotranspiration changed less than 10%. In 2022, human contributions accounted for 30% of statewide evapotranspiration and 80% in managed lands. In 2023, natural evapotranspiration increased, and human contributions fell by 30%, yet still comprised nearly 50% of evapotranspiration in managed areas. Our findings underscore the enduring role of human activity on California's hydrology, even during wet years, and demonstrate a framework to separate natural and anthropogenic controls on evapotranspiration.

Climate-related threats to California's land and ecosystems present an urgent concern for the future economic and food security of the United States. California's working landscapes (e.g., land dedicated to agriculture, forestry, and outdoor recreation, among others) generate more than $333 billion in sales, $85 billion in earnings, and produce more than 1.5 million jobs[1]. These earnings do not include the additional benefits provided by ecosystem services such as carbon sequestration, air quality regulation, erosion regulation, etc. The largest contributor to working landscape earnings in California is the agricultural sector, which generated more than $50 billion in annual revenue in 2022, predominantly from irrigated crops[2]. Over a third of the vegetable and over three-quarters of the fruit and nut production for the United States comes from California[3]. Critically, water is necessary to support working landscapes for irrigation, forest growth, and overall ecosystem health[4–6]. California's changing environmental conditions, including increased frequency and severity of drought[7,8], and more extreme swings from dry to wet conditions[9,10] pose a threat to California's water resources, thereby threatening these working landscapes and national security.

To predict future water resources and water security, we need to understand how California's changing climate system will impact different components of the water budget, including precipitation, water flow into and out of watersheds, water storage, and evapotranspiration (ET, the sum of evaporation from soils and surface-water bodies and plant transpiration)[11]. Of these components, ET is particularly important, as it is the second-largest component of the water budget. In California, about 60% of the total precipitation returns to the atmosphere as ET[12]. ET often approximates consumptive water use (i.e., all the water in a system that cannot be recovered or reused, including water consumed by plants or humans and evaporated). ET is therefore a critical component of sustainable water resources decision making, particularly in agriculture[13–17], and an important indicator of vegetation stress[18,19].

ET is controlled by environmental factors, including temperature, soil moisture, solar radiation, wind, and atmospheric vapor pressure[20–22]; biotic factors like ecosystem type, vegetation health, and plant functional traits[23–25]; and water applied via irrigation[26,27]. Accordingly, ET is a function of both natural (environmental controls and vegetation health) and human (management decisions such as forest thinning or crop selection, and irrigation practices) contributions. Similarly, the water source used to support ET can come from both natural and human sources—termed "green" and "blue" water. Green water is precipitation that does not run off and temporarily contributes to soil water storage. Blue water is surface and groundwater stored in rivers, lakes, aquifers, and dams, and can be extracted

[1]NASA Jet Propulsion Laboratory, California Institute of Technology, Pasadena, CA, USA. [2]Lamont-Doherty Earth Observatory of Columbia University, Palisades, NY, USA. [3]Bren School of Environmental Science & Management, University of California, Santa Barbara, Santa Barbara, CA, USA. [4]Department of Environmental Science Policy and Management, University of California Berkeley, Berkeley, CA, USA. ✉e-mail: zoe.a.pierrat@nasa.jpl.gov; christine.m.lee@nasa.jpl.gov

for human use[28,29]. The relative contributions of green and blue water to ET are therefore important for anticipating and managing future changes to the water budget and future water resources. Importantly, both the amount of water available to support ET from both green and blue water sources, as well as the natural and environmental controls on ET, are sensitive to the rapidly changing environmental conditions in California.

Climatically, seasonal and interannual swings from dry to wet conditions have amplified in the western US[30,31]. However, the subsequent impacts of these types of swings from extreme dry to extreme wet conditions on ET and the total water budget in California are unknown. This is in part because ET depends on vegetation's natural responses, which are regionally specific, and human intervention, which can vary greatly according to local and state policy[32]. Consequently, there is no consensus over whether ET will increase or decrease under drought, and the ET response is highly dependent on the strength and duration of the drought and policies around water management.

In agricultural regions under drought, higher evaporative demand leads to increased ET[33] and decreased soil moisture (i.e., less green water available). This results in an increased need for irrigation. Increased irrigation can lead to a further amplification of ET and an accelerated depletion of groundwater storage[7]. In this case, blue water contributes more to ET[34]. In contrast, water use regulations (such as those imposed in California in 2022) can lead to a reduction in ET due to curtailments on irrigation. This can result in compromised agricultural activity due to high vegetation stress[35].

Non-agricultural regions may also experience increased ET as a result of higher evaporative demand[33]. On the other hand, ET can also be reduced under extreme drought, driven by a decline in available water sources for soil evaporation and plant transpiration, as well as vegetation mortality[36,37]. In forests, forest thinning has been proposed as an effective drought stress management decision, as it has been shown to increase forest resiliency under drought[38]. The subsequent impacts of thinning on ET show varied results[39]. Prior work has reported both a net decrease in ET as a result of the reduction in tree cover[40]; and no change or an increase in ET as a result of higher individual tree transpiration compensating for the loss in tree area, and higher wind speed and solar radiation on the forest floor[41–43]. These types of management decisions impact ET by modifying green water movement. Enhanced precipitation and total water storage anomalies (i.e., more green water available) following drought can also improve vegetation recovery and rebalancing of ET[33,44]. A vegetation boom following high precipitation will increase ET but may also contribute to an enhanced risk of wildfire, further altering the water budget[45–47].

Changes to ET under drought conditions are therefore highly uncertain, and depend on the duration, severity, and intensity of the drought/dry period, and human management decisions. Understanding how natural and human contributions impact the ET response under climatic swings in California, which are expected to become more extreme and more frequent, will improve our understanding of the regional water balance, including groundwater recharge and future water resource availability[48]. Here, we compare how vegetation and human processes in an exceptionally wet year (2023) and an exceptionally dry year (2022) contributed to changes in ET and subsequent water demand in California. Specifically, we ask, (1) how did ET change in California from an extreme dry year (2022) to a record-breaking water year (2023)? and (2) how did the relative contributions of natural and human influences on ET contribute to or mitigate those changes?

To answer these questions, we use a combination of observationally constrained and modeled ET to disentangle natural and human impacts on ET. We focus our study on just summer months (May–September) to remove phenological effects from this analysis. We obtain *total* ET (i.e., observed ET as a result of both natural and human controls) using observationally constrained estimates of ET from OpenET[49]. We obtain *natural* ET using modeled ET from the National Land Data Assimilation System (NLDAS)[50,51]. NLDAS models ET based on the natural processes governing the hydrologic cycle, including soil infiltration, surface runoff, drainage, and

atmospheric and vegetation controls on ET. NLDAS does not include potential human influences on the water budget, including irrigation, management decisions, or human groundwater extraction. Therefore, we assume the difference between these two datasets (OpenET-NLDAS ET) can be attributed to human controls on ET, including irrigation and management decisions[27] and the fractional human impact (FHI) on ET can be quantified as the ratio of human ET to total ET. This approach is applied across the entire state of California, regardless of vegetation class, to assess both the uncertainty in datasets used[51–53] and potential human impacts on ET in California's working landscapes beyond the agricultural sector[34]. Our approach does not rely on reported crop characteristics or irrigation and does not make assumptions about assumed growing areas or crop-specific inputs, which can hinder hydrologic modeling of green and blue water[54–57]. This enables us to distinguish the human impact on the water cycle across a range of vegetation types, which is critical for efficient water resource management[34].

This study has implications for sustainable water management in California, which has the 4th largest economy in the world as of 2025 and is highly dependent on the success of its working landscapes[58]. Using these cutting-edge data and modeling approaches, we shed light on how human consumptive water use and overall water resources may shift under a changing climate. Our methodological approach for separating natural and human controls on ET has potential for understanding future changes to ET across the broader US, and eventually water-limited regions globally.

## Results

California is dominated by shrub/scrub (45% of regridded pixels), evergreen forest (17%), grassland/herbaceous (14%), cultivated crops (10%), barren land (3%) and developed, medium intensity land (2%) (Fig. 1a). Growing season baseline (2016–2021) ET data from OpenET (total ET), NLDAS (natural ET), and the difference between them (human ET), reveal the unique spatial patterns and variable contributions to ET across California (Fig. 1b–d). Growing season baseline (2016–2021) total ET is an average of 45 with a standard deviation of $\pm 29$ mm month$^{-1}$ across the state. Spatially, the majority of ET is from northern and coastal California (wherever evergreen forests dominate) and the Central Valley (primarily cultivated crops). By land cover type, the highest average growing season baseline (2016–2021) total ET (Fig. 1b) is found in evergreen forests ($79 \pm 18$ mm month$^{-1}$), followed by cultivated crops ($69 \pm 18$ mm month$^{-1}$), developed, medium intensity land ($60 \pm 11$ mm month$^{-1}$), grassland/herbaceous ($34 \pm 19$ mm month$^{-1}$), shrub/scrub ($33 \pm 22$ mm month$^{-1}$), and barren land ($14 \pm 11$ mm month$^{-1}$).

Natural ET comprises the majority of total ET across the state, with an average growing season baseline of $32 \pm 21$ mm month$^{-1}$ (71% of total ET) (Fig. 1c). The highest natural ET comes from evergreen forests ($60 \pm 11$ mm month$^{-1}$), followed by grassland/herbaceous ($36 \pm 18$ mm month$^{-1}$), shrub/scrub ($25 \pm 17$ mm month$^{-1}$), cultivated crops ($19 \pm 11$ mm month$^{-1}$), developed, medium intensity ($16 \pm 5$ mm month$^{-1}$), and barren land ($13 \pm 15$ mm month$^{-1}$). Human ET has an average growing season baseline of $13 \pm 23$ mm month$^{-1}$ (29% of total ET) (Fig. 1d). The highest average growing season baseline human ET comes from cultivated crops ($49 \pm 20$ mm month$^{-1}$), followed by developed, medium intensity ($43 \pm 15$ mm month$^{-1}$), evergreen forests ($19 \pm 18$ mm month$^{-1}$), shrub/scrub ($7 \pm 18$ mm month$^{-1}$), barren land ($1 \pm 11$ mm month$^{-1}$), and grassland/herbaceous ($-2 \pm 16$ mm month$^{-1}$). Notably, the observed standard deviation for each land cover classification is less than the expected RMSE based on previous work quantifying uncertainty by land cover type in these datasets[51–53] and propagated through our analysis (see "Methods," Table 1). The lower standard deviations give us confidence that additional error introduced by the aggregation of multiple land cover classes within a single pixel is insignificant compared with the inherent uncertainty in the data themselves. Furthermore, for land cover classes that we would not expect to have a human influence on ET (shrub/scrub, barren land, grassland/herbaceous), human ET is not statistically

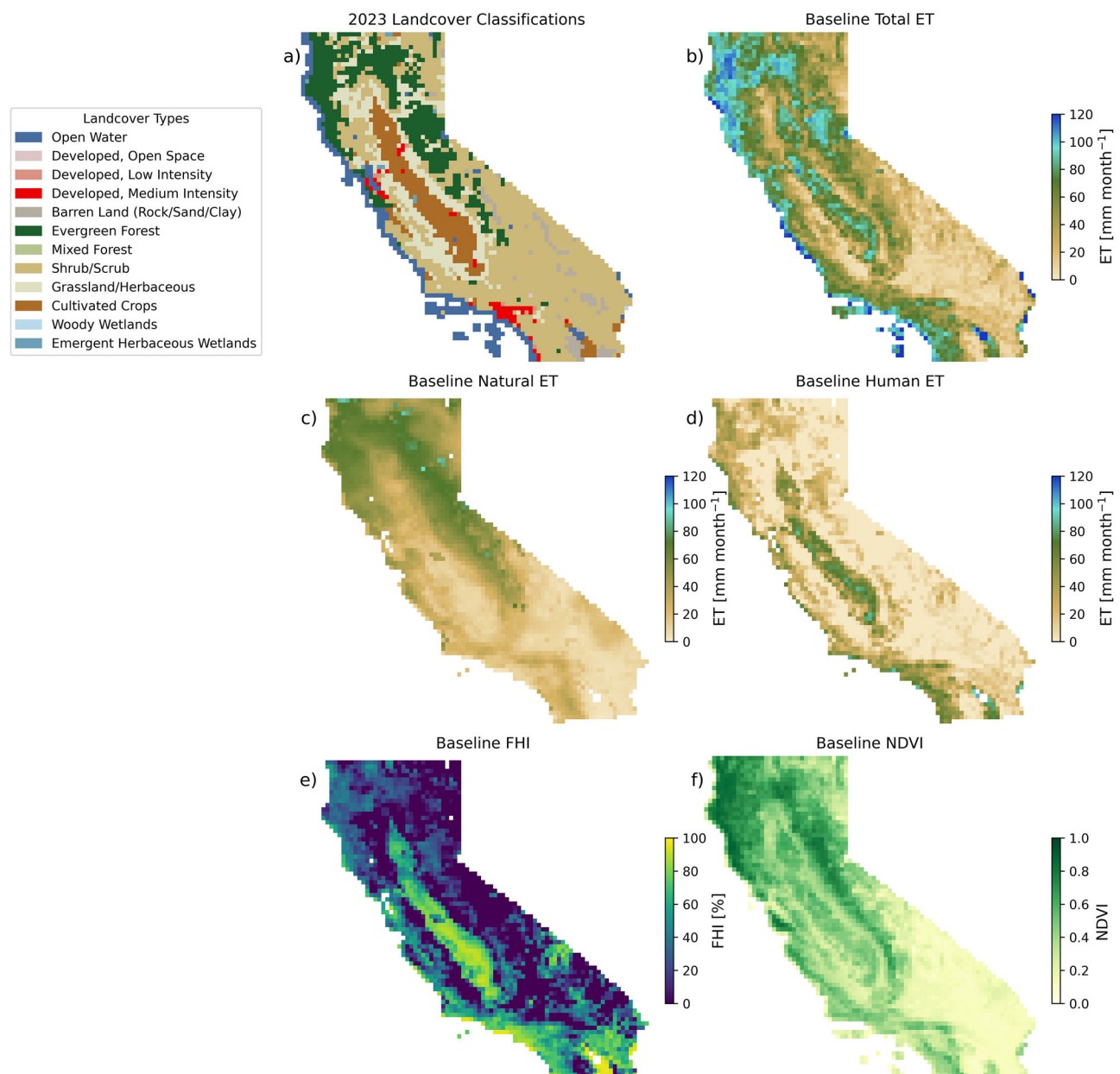

**Fig. 1 | Growing season baseline (2016–2021) conditions across California. a** Land cover classifications across California for 2023. The summer (May–September) baseline (2016–2021) ET for (**b**) total ET from OpenET, (**c**) natural ET from NLDAS, (**d**) human ET as the difference between OpenET and NLDAS ET, as well as (**e**) FHI, and (**f**) NDVI. All pixels were aggregated to 0.125 × 0.125° resolution to match the NLDAS natural ET dataset.

different from zero (i.e., human ET = 0 is within the average ± the standard deviation of reported values). The one exception to this is in evergreen needleleaf forests, which could have some human influence because of forest thinning[39,41,42] or spatial aggregation across cropland and developed land[27]. The expected uncertainty based on the OpenET and NLDAS datasets in evergreen needleleaf forests is 30 mm month$^{-1}$, suggesting that although human ET in evergreen forests has a high bias (more on this in section "Discussion"), it is not statistically different from zero.

Baseline FHI is highest in the Central Valley and along southern coastal California, where cultivated crops and developed land dominate and humans have a strong influence on ET (Fig. 1e). In these managed areas, developed land has the highest FHI at 71 ± 16%, closely followed by cultivated crops with a FHI of 70 ± 19%. Baseline NDVI reflects the state of vegetation cover across California (Fig. 1f). Baseline growing season NDVI is highest in evergreen forests (0.67 ± 0.12, unitless),

followed by cultivated crops (0.46 ± 0.09), grassland/herbaceous (0.42 ± 0.15), developed, medium intensity (0.31 ± 0.06), shrub/scrub (0.30 ± 0.18), and barren land (0.12 ± 0.06). Taken together, these results highlight the strong impact of human processes on total ET across California, with human contributions to baseline ET are comparable or greater than natural ET across much of the state.

In 2023, California was hit by repeated Pacific atmospheric rivers bringing extreme precipitation, snowfall, and increased snowpack levels compared with the historical baseline, creating a "record-breaking water year"[59]. This record water year was preceded in 2022 by some of the most extreme drought conditions in California history. Precipitation data across the state are consistent with these previous findings (Fig. 2).

Comparing total, natural, and human average ET in the growing season in 2022 and 2023 clarifies the impact of climatic swings on ET across California. Despite being an exceptionally dry year in 2022, California experienced an average total ET of 49 ± 28 mm month$^{-1}$ (Fig. 3a),

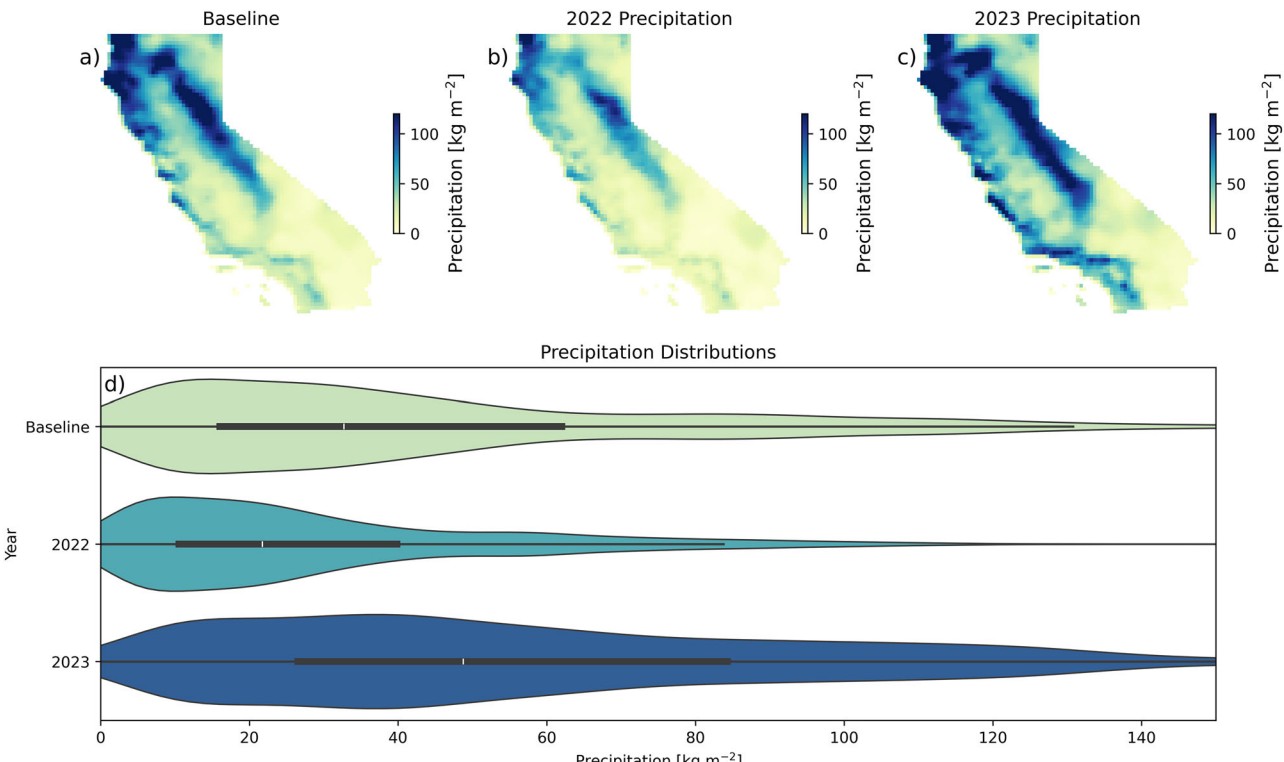

**Fig. 2 | Annual average precipitation across California.** Precipitation for **a** the baseline (by-pixel average, 2016–2021) years, (**b**) 2022, and (**c**) 2023. (**d**) Shows the distribution of precipitation values across the state, highlighting the extreme drought year in 2022 and the "record-breaking water year" in 2023. White dots represent the median, and darker regions are the inner quantile range.

or an average increase in ET of 16% relative to the 2016–2021 baseline. This is consistent with previous work suggesting an increase in ET in California despite decreased precipitation in both natural[60] and agricultural[61] land. In 2023, the exceptionally wet year, California experienced an average total ET of $52 \pm 26$ mm month$^{-1}$, a 35% increase relative to the 2016–2021 baseline (Fig. 3b). From 2022 to 2023, California experienced a marginal average increase in ET by $3 \pm 9$ mm month$^{-1}$ (6% increase from 2022) (Fig. 3c), despite the large increase in precipitation. Changes in ET were highly non-uniform, and mediated by large changes to natural vs human contributions to ET. Natural ET contributed an average of $29 \pm 1$ mm month$^{-1}$ (59% of total ET) in 2022 (Fig. 3d), while in 2023, natural ET contributed an average of $47 \pm 23$ mm month$^{-1}$ (90% of total ET) across the state (Fig. 3e). Between 2022 and 2023, natural ET increased by an average of $17 \pm 13$ mm month$^{-1}$ (Fig. 3f). Human ET contributed an average of $20 \pm 24$ mm month$^{-1}$ (41% of total ET) in 2022 (Fig. 3g) and $5 \pm 21$ mm month$^{-1}$ (10% of total ET) in 2023 (Fig. 3h). Between 2022 and 2023, human ET decreased by an average of $-14 \pm 14$ mm month$^{-1}$ (Fig. 3i). FHI in 2022 (Fig. 3j) and 2023 (Fig. 3k) was highest in the Central Valley and southern California and exhibited an average decrease across the state of $-31 \pm 39\%$ from 2022 to 2023 (Fig. 3l). Despite this overall decrease, a high FHI impact remained in the Central Valley and Los Angeles in 2023. Average NDVI was $0.38 \pm 0.20$ in 2022 (Fig. 3m) and $0.41 \pm 0.20$ in 2023 (Fig. 3n), representing an average increase of 0.03 (8%) (Fig. 3o).

The changes in ET and ultimate impacts to the water balance become clearer when breaking up changes in ET, FHI, and NDVI by land cover classification (Fig. 4). Between 2022 and 2023, natural land cover classifications (shrub/scrub, grassland/herbaceous, evergreen forests) showed small but statistically significant changes to ET, while managed lands (developed land, cultivated crops) showed no significant change in ET (Fig. 4a). Natural ET, which shows a clear increase in ET in 2023 across the board: shrub/scrub ($23 \pm 16$ mm month$^{-1}$ in 2022 to $39 \pm 21$ mm month$^{-1}$ in 2023), grasslands ($29 \pm 22$ mm month$^{-1}$ in 2022 to $54 \pm 17$ mm month$^{-1}$

in 2023), developed land ($12 \pm 6$ mm month$^{-1}$ in 2022 to $30 \pm 9$ mm month$^{-1}$ in 2023), and crops ($12 \pm 9$ mm month$^{-1}$ in 2022 to $37 \pm 15$ mm month$^{-1}$ in 2023) (Fig. 4b). Human ET shows a smaller change between years for land cover types which are not expected to have a human impact (barren land, shrub/scrub, grasslands, evergreen forests) while developed land and cultivated crops show a clear decrease in human ET in 2023 ($44 \pm 12$ mm month$^{-1}$ in 2022 to $25 \pm 16$ mm month$^{-1}$ in 2023 for developed land; and $58 \pm 19$ mm month$^{-1}$ in 2022 to $36 \pm 21$ mm month$^{-1}$ in 2023 for cultivated crops) (Fig. 4c). Consequently, the FHI on ET changed significantly for developed land ($78 \pm 12\%$ in 2022 to $41 \pm 25\%$ in 2023) and cultivated crops ($80 \pm 15\%$ in 2022 to $47 \pm 27\%$ in 2023). NDVI shows small changes between the two years, with the largest increases in NDVI occurring in grassland/herbaceous ($0.37 \pm 0.09$ in 2022 to $0.42 \pm 0.09$ in 2023), followed by cultivated crops ($0.45 \pm 0.09$ in 2022 to $0.48 \pm 0.09$ in 2023).

## Discussion
With this work, we clarify how ET changed in California during a "record-breaking water year" (2023)[59] following a record dry year (2022) and how the relative contributions of natural and human influences on ET mediated those changes. In the drought year (2022), statewide total ET showed an increase relative to the historic baseline as a result of high temperatures and increased evaporative demand[33,61–64]. In the water year (2023), statewide total ET showed only a small increase (average <10% change), despite the large increase in precipitation (141% of the statewide average[65]). The stability of ET is explained by the large role humans had on ET in 2022, which was reduced in 2023 when increased precipitation made more green water sources available. Our results highlight how changing contributions of natural and human processes mitigate changes in ET due to climatic swings, and demonstrate the environmental (temperature, humidity, natural disturbances such as fires or pests) and human controls (irrigation, crop type, forest management) on ET[24,25,46,66,67].

In agricultural and developed land, drought conditions in 2022 increased the need for irrigation[68,69] and a likely depletion of groundwater

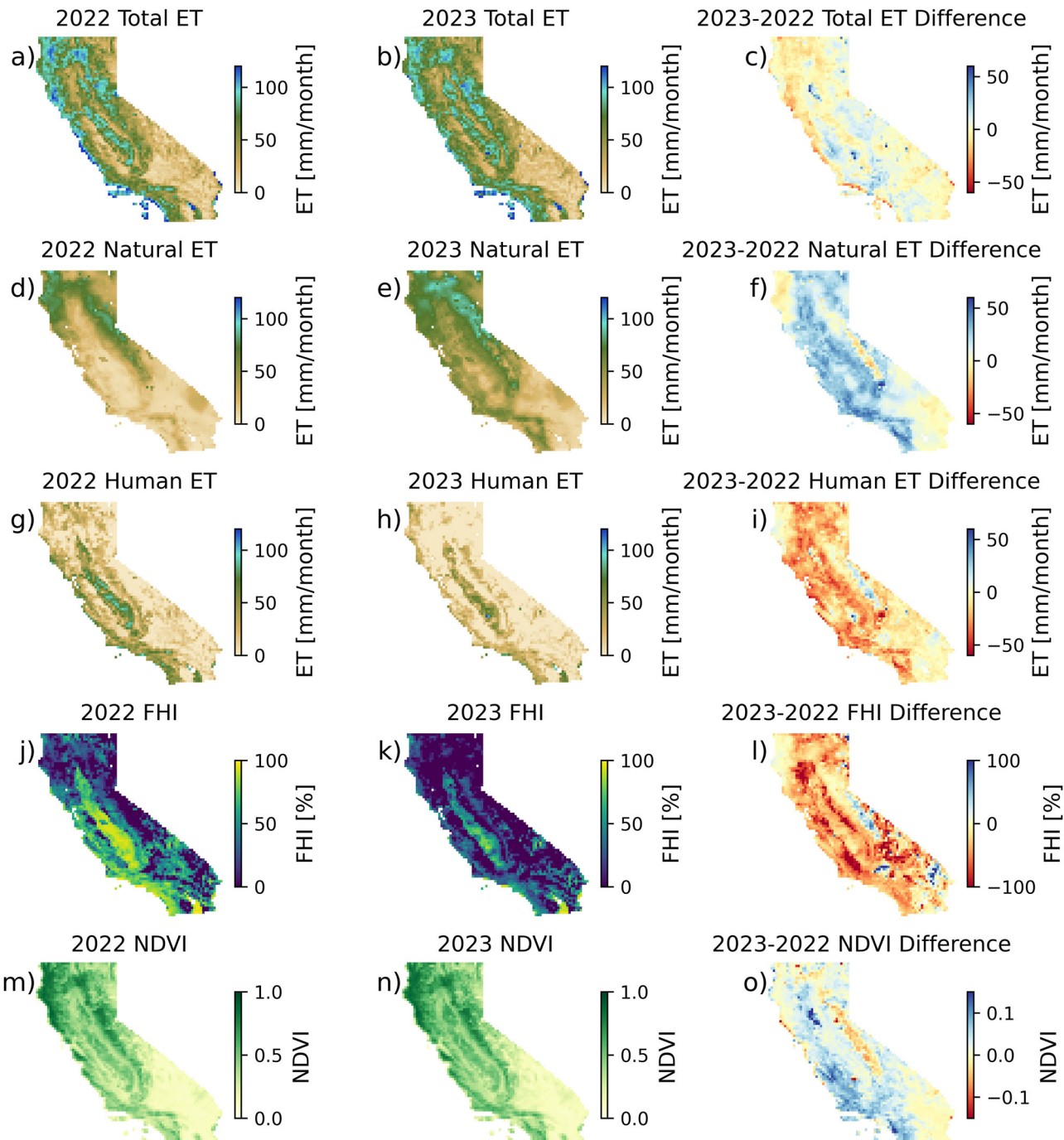

**Fig. 3 | Changes in ET as a result of climatic swings.** Growing season (May–September) ET in the dry year (2022, left column), the wet year (2023, middle column), and the difference between them (right column) for (**a–c**) total ET from OpenET, (**d–f**) natural contributions to ET from NLDAS, (**g–i**) human contributions to ET as the difference between OpenET and NLDAS ET, (**j–l**) the FHI on ET, and (**m–o**) NDVI.

storage[7] as more water blue water sources are required. These conditions were present across California from 2020 to 2022[70], which led to the high FHI (80% in agricultural and developed land) in the dry year (2022) and the prevention of changes to total ET in the wet year (2023). Given the long-term drought in California prior to 2022, the intensity of human ET may also be driven by exacerbated water stress from the lagged effects of drought. Despite a reduction in human ET in 2023, the FHI on managed lands (developed land and cultivated crops) remained high (nearly 50%), suggesting that future "boom water years" will still be insufficient for the water demands of California. The high FHI in California's agricultural land can in part be attributed to the fact that most of the precipitation in California

occurs during winter[47,71], but the growing season for most crops is during the summer[72–74], necessitating the use of irrigation to provide sufficient soil moisture. Drought conditions are expected to increase through the 21st century, which further underscores the need for water-smart agricultural practices[26,75].

There are several approaches for reducing the human-driven contribution to total ET. Recent work[26] has demonstrated a potential 10% reduction in water consumption because of (1) crop selection (from high-ET crops to lower ET-crops), (2) farming practices (e.g., reduced irrigation to the same crops), or (3) fallowing 5% of the land. This research revealed a maximum 94% reduction in agricultural ET as a

**Fig. 4 | Evaluation of changes in ET based on land cover classifications.** The distribution of ET broken up by land cover and separated by year (dry 2022 vs wet 2023) for (**a**) total ET, (**b**) natural ET, (**c**) human ET, (**d**) FHI, (**e**) NDVI. The red stars (*) indicate groupings where there was a statistically significant difference ($p < 0.05$ in a Mann–Whitney U-test). Dashed lines within the violin plots represent the median and inner quantile range. Vertical black lines for total ET, natural ET, and human ET indicate the anticipated range of values as a result of inherent uncertainty in the datasets used. This is quantified using RMSEs reported in prior work and propagated through calculations in this study as described in the section "Data Processing" and reported in Table 1.

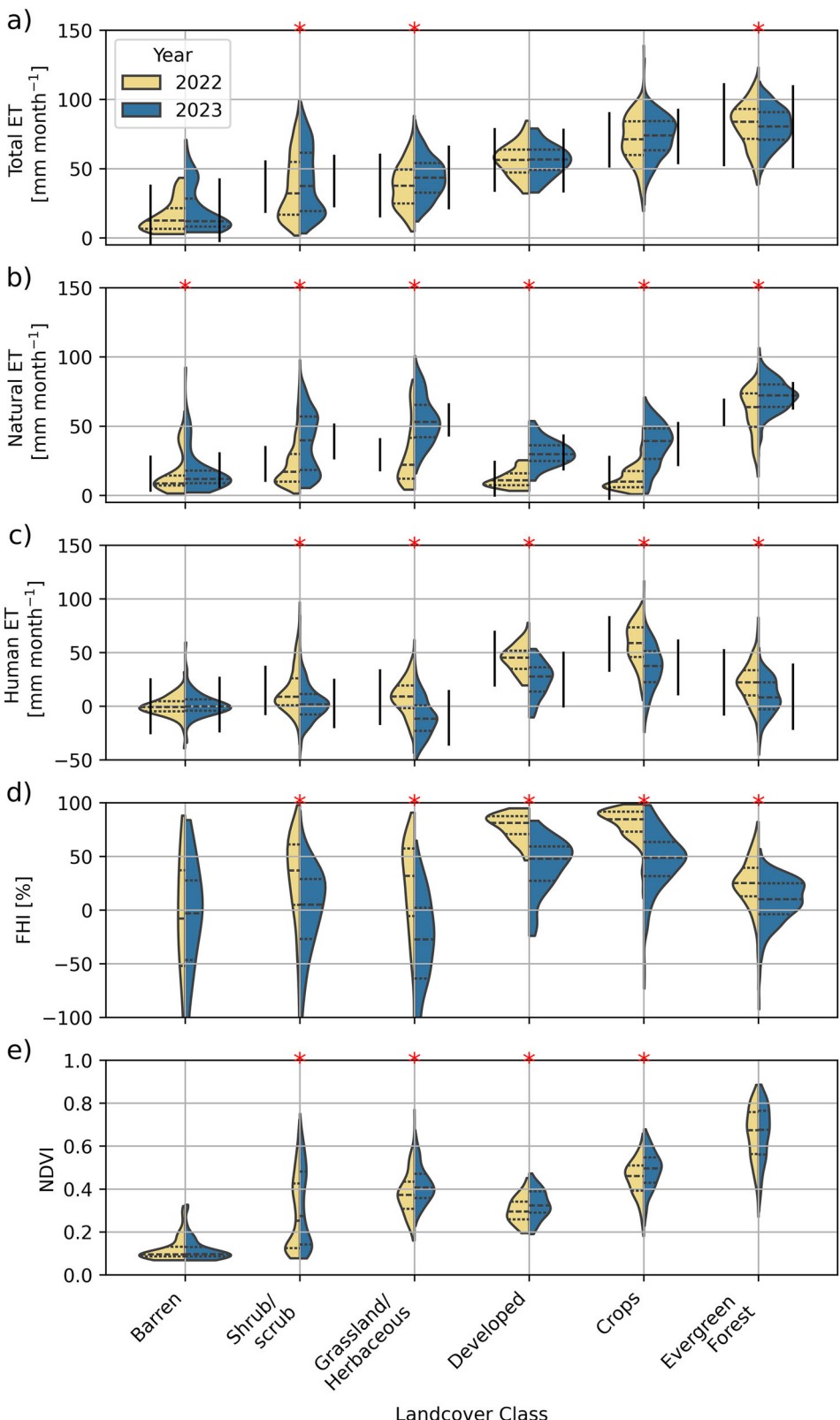

result of switching to the lowest water use crops, although monolithic production of a single crop is unlikely to meet agricultural demand. Water rights and water outlooks[76] affect which annual crops get planted, and determine ET in agricultural lands as more water-intensive crops will lead to higher ET[26,77,78]. Our results show an increase in NDVI in cultivated crops in 2023, which may reflect a change in planted crops to more water-demanding crops[79] following a high water outlook for 2023[2,80–82], a

decrease in the amount of fallowed land[26,83], or plants under less water stress. Our results highlight that water-smart management strategies will be important even in years with high water outlooks, and water rights and water outlooks[76,84] should reflect these concerns.

In shrub/scrub, barren land, grassland/herbaceous, and evergreen forests, humans did not have a statistically significant impact on ET. This suggests that any management decisions that may impact ET (such as forest

thinning) are currently inconsequential to the state water budget, and natural ecosystems are primarily responding to natural controls. Although management decisions did not result in changes in ET at the spatial scale of this study, these decisions may still have an impact on local water budgets by modifying green water stored in the soil and moved through plants. In natural areas, increased green water availability, like that observed in 2023, can increase ET due to higher vegetation health and abundance. Grasslands, barren land, and shrub/scrubland all showed subtle increases in NDVI due to higher water availability in 2023. This led to strong increases in ET in these areas as transpiration increased[20]. In contrast, evergreen forests showed smaller changes in NDVI and ET between 2022 and 2023. NDVI is a poor predictor of productivity in evergreen forests[85]; therefore, NDVI may not accurately capture interannual variability in these ecosystems. The fact that ET also remained relatively consistent in evergreen forests between years suggests a higher resilience to extreme swings in precipitation from year to year. The stability in ET in evergreen forests is likely because evergreen forests are able to access deeper groundwater stores, making groundwater recharge in these areas following extreme precipitation critical for the sustained resilience of these ecosystems[44]. Multi-year droughts can deplete deep groundwater reserves and lead to forest die-off[62], emphasizing the need for groundwater recharge during high water years.

The effect of hydroclimate volatility on vegetation and ET in California, as observed in 2022 and 2023, will have cascading consequences for the natural ecohydrology of California and wildfire regimes[10,47]. When a wet year follows a dry year, the increased water and nutrient availability can lead to a boom in vegetation growth, particularly in Mediterranean climates and semi-arid shrubland ecosystems[86], consistent with what was observed in this study. This boom in vegetation growth increases fuel loading and subsequently, increases wildfire severity[87–89]. Climate change is amplifying this wet-to-dry trend by intensifying both jet streams and strong winds, which can ignite and spread fire[90]. This pattern was observed in the severe California wildfire season in 2020[91], following 2019 being the first non-drought year in California since 2011[92]. The 2024 fire season initially followed this trend, burning more acreage than the following 5 year-average in the early summer months[93], and may have contributed to the extreme fire events in Los Angeles in early 2025. Future research should investigate the connections of 2023 precipitation patterns to the 2024–2025 fire season.

The approach used in this study to delineate total, natural, and human ET processes has the potential to be used in other regions and across years to understand the water balance and future water resources. Despite this potential, there are several limitations to this work, leading to potential errors in quantifying the relative contributions of human ET. These errors appear most notably as reported negative human ET and FHI in barren land, shrub/scrub, and grasslands, as well as a non-zero FHI for evergreen needleleaf forests (Fig. 4). We attribute these issues to two main limitations. First, known biases in the models themselves as a consequence of model forcing data errors, model structure deficiencies, and model calibration errors[51,52]. OpenET has higher errors in shrublands and grasslands compared with croplands[52]. Despite this variability, OpenET does not show a bias in shrub and grassland ecosystems[52], giving us confidence that, despite the large scatter in ET values, the statewide average will converge on the true value. This is consistent with our results, which show an average of 0 mm month$^{-1}$ human ET and 0% FHI in barren land, shrub/scrub, and grasslands, despite notable spread in the data. In evergreen needleleaf forests, OpenET has a known high bias[52], and NLDAS has a low bias[51]. Taken together, this leads to an over-inflation of human ET in evergreen needleleaf systems, consistent with our results. Second, the coarse resolution of NLDAS requires spatial aggregation of multiple land cover types and may not adequately capture fine-scale variability in natural and human ET impacts present in the OpenET data. This spatial aggregation is particularly relevant when considering fine-scale changes in agricultural regions such as crop rotation and fallowing. Future work should

consider potential downscaling approaches for understanding more water-efficient irrigation practices[26]. The aggregation of multiple land cover types in a single pixel may also contribute to the non-zero FHI in areas that would not be expected to have an irrigation/human influence (e.g., evergreen forests) if irrigated regions are within the pixel. However, the spread in the data was less than the expected RMSE based on previous work, suggesting that the spatial aggregation is less important than improving the data products themselves.

These known limitations do not undermine our main finding that dry–wet climatic swings do not significantly change total ET in California due to the strong human signature on ET, particularly in managed lands. Our results show significance despite known uncertainties in data sources, as the changes in human ET and FHI in managed lands are statistically significant and notably larger than the spread in the data (Table 1 and Fig. 4). Our approach demonstrates the ability to capture changing contributions to ET from human and natural processes as a consequence of extreme climatic swings. Delineating human and natural controls on ET will be increasingly important for managing blue and green water sources now and into the future. As irrigation expands into water-stressed regions globally[94,95], and water stress is expected to increase[96], the downstream consequences of these changes to the entire water balance are largely unknown. By quantifying the human contribution to ET, we can better understand how human management decisions impact the entire water budget and the interplay between humans and natural climate influence. Prior work has demonstrated a decrease in groundwater storage during megadroughts[7], but direct attribution of this depletion to human or natural controls can only be made by quantifying the human component of ET. Our approach demonstrated here can better link components and drivers of the water budget, including groundwater extraction and recharge, paving the way for understanding future water availability and guiding water resource management.

## Conclusions
Atmospheric rivers during the winter and spring of 2023 led to record-breaking precipitation across the state of California following an extreme drought year in 2022. We investigated the differences in ET between an exceptionally dry year (2022) and an exceptionally wet year (2023) to better understand and quantify human activities on the water cycle of California. Novel remote sensing and modeling data sets from OpenET and NLDAS enabled us to partition total observed ET into natural and human components and investigate how human influence is mitigating changes in ET during climatic swings in water availability. We found that total ET only slightly increased in the wet year, largely because of an exceptionally high human contribution to ET during the dry year. During the wet year, natural ET is significantly increased in both managed and unmanaged areas, allowing the human influence on ET to significantly decrease. Despite this, the human signature in managed lands (cultivated crops and developed land) remained nearly 50%, underscoring the extreme overdraw of groundwater to support California's agricultural industry. These results show statistical significance notwithstanding uncertainties in the OpenET and NLDAS datasets. In a changing climate, where these types of swings are expected to be more extreme and more frequent, our results highlight why high-water years cannot alone restore California groundwater resources, and why understanding the local factors impacting ET response during climatic swings will help humans make smarter water decisions.

## Methods
To quantify the differences between ET responses to climatic swings from exceptionally dry to exceptionally wet water years, we used a combination of datasets on land cover, vegetation health, and ET. The years 2022–2023 in California provide an ideal natural case study. These two years allow us to test the response of vegetation to extreme interannual swings while minimizing the impact of land cover change between years.

**Table 1 | Reported RMSE values from previous studies and propagated RMSE values for this study.**

| Model<br>Source | OpenET<br>52 | VIC<br>53 | MOSAIC<br>53 | NLDAS<br>Propagated from VIC and Mosaic | Human<br>Propagated from OpenET and NLDAS |
|---|---|---|---|---|---|
| Grassland/herbaceous | 23 | 14 | 20 | 12 | 26 |
| Shrub/scrub | 19 | 15* | 20* | 13 | 23 |
| Barren land (Rock/sand/clay) | 23* | 15* | 20* | 13 | 26 |
| Evergreen forest | 30 | 7 | 18 | 10 | 31 |
| Cultivated crops | 20 | 23 | 22 | 16 | 26 |
| Developed, medium intensity | 23* | 15* | 20* | 13 | 26 |

*Indicates that RMSE was not provided for this land cover type in previous studies, so the mean RMSE across all land cover types was used. All values are reported in mm month$^{-1}$.

## Datasets

To evaluate how changes in precipitation and available water impact different land cover types, we used land cover classifications from the National Land Cover Database (NLCD) (United States Geological Survey, 2023). NLCD data are available over the conterminous United States from 1985 to 2023 at a 30-m resolution. Open water classes were excluded from this analysis but are still included in maps for visualization. Open water accounts for 8% of land cover in California and is dominated by coastal water.

To quantify *total* observed ET across the state of California, we used monthly ET data from OpenET[49]. OpenET data are available at a 30 m × 30 m resolution from 2016 to present. OpenET uses satellite-based land surface temperature and meteorological data from Landsat, Terra/Aqua MODIS, Suomi NPP VIIRS, GOES, and Sentinel-2[49,52] to drive 6 different ET models. The models included in OpenET include both energy balance and reflectance-based approaches to ET mapping[49]. These approaches directly model the physical processes governing ET using satellite-based observations of the evaporative response. Therefore, the products included in OpenET approximate total observed ET, including both natural and human contributions to ET, and are independent of blue or green water sources. OpenET data have been validated against a benchmark eddy flux ET dataset, which includes a number of sites in California's Central Valley[97]. Validation of OpenET data shows good performance across vegetation types at a monthly resolution, particularly across cropland sites ($R^2 = 0.9$, Table 1 for RMSE)[52]. In this study, we use the OpenET ensemble estimate, which has better performance than any individual ET model alone[52].

To assess the *natural* contributions to ET (not including contributions due to irrigation), we used monthly modeled ET data from the National Land Data Assimilation System (NLDAS), which are provided at a 0.125 × 0.125-degree resolution (approximately 150 km$^2$ pixels) from 1979 to present[50]. NLDAS uses gauge-based precipitation, downward shortwave and longwave radiation, and surface meteorology reanalysis (10-m wind-speed, 2-m air temperature, 2-m specific humidity, surface pressure) to drive three land surface models to produce outputs of naturally occurring surface fluxes (including ET), soil moisture, snow cover, and streamflow[50]. Three models within NLDAS (Noah[98], Mosaic[99], and VIC[100]) simulate the surface energy balance using the meteorological forcing data. Importantly, NLDAS models the natural processes contributing to ET, which do not include changes in ET based on human influence (e.g., irrigation)[27]. Unlike OpenET, NLDAS does not include a land surface temperature constraint, which would help distinguish natural ET based on the surface energy balance from total ET (including processes not considered in the model)[50]. Validation of NLDAS models has been performed using spatial and temporal averaging over AmeriFlux stations to overcome spatial-scale incompatibility issues[51,53]. NLDAS demonstrates good performance representing ET across vegetation types[51], particularly over natural ecosystems with no known irrigation. Therefore, NLDAS does a good job of representing naturally occurring ET fluxes. Prior validation work has shown that the lack of explicit representation of irrigation is a key source of error in the NLDAS models[50], making it unsuitable for representing total ET over irrigated areas. After evaluating the ability of each of the individual NLDAS land surface models

to capture ET in our research domain using OpenET point data over non-irrigated regions, we averaged the VIC[100] and Mosaic[99] models as the highest performing combination, given Noah's low predictive power over California[51].

We inferred *human* impact on ET by taking the difference between OpenET (i.e., total ET) and NLDAS ET (i.e., natural ET)[27]:

$$\text{Human ET} = \text{Total ET} - \text{Natural ET} \qquad (1)$$

Under this definition, human ET reflects all processes impacting the ET signal that are not quantified in the NLDAS models, as well as uncertainty in both datasets. By calculating human ET over land cover classes that are not expected to have a human influence on ET (e.g., evergreen needleleaf forests), we are able to (1) provide a statistical baseline reference for our results (i.e., do natural landcover types show human ET = 0); and (2) account for human ET in agricultural and developed land that might go unaccounted for in the spatial aggregation of heterogeneous land cover classes within the coarse NLDAS pixel (Data Processing section).

We define FHI as:

$$\text{FHI} = \frac{\text{Human ET}}{\text{Total ET}} \times 100\% \qquad (2)$$

To assess overall changes in vegetation coverage as a key driver of ET, we used normalized difference vegetation index (NDVI) data from the NASA Visible Infrared Imaging Radiometer Suite (VIIRS) Land Program[101,102]. Data are provided monthly on a 1 km × 1 km grid and extend back to January 2012.

To highlight the differences in precipitation across the state in 2022 and 2023 compared to the baseline, we used total precipitation data from NLDAS secondary forcing data[50,103]. We averaged monthly data for the baseline years (2016–2021), 2022, and 2023 across the full year.

## Data processing

To evaluate uncertainty in this analysis, we consider both the spatial variability in the datasets as observed in this analysis and previously reported uncertainty. Both NLDAS and OpenET datasets have inherent uncertainty, which has been thoroughly evaluated in previous work[51–53]. To assess the robustness of our findings, we compared the standard deviation of our data to the root mean squared errors (RMSEs) reported in prior work (Table 1). For OpenET, Volk et al.[52] provide RMSEs in mm month$^{-1}$ broken up by land cover class. For NLDAS, Xia et al.[51] provide initial RMSEs for individual models in NLDAS. This work was expanded by ref. 53, which further evaluated NLDAS RMSEs across the United States in mm month$^{-1}$ broken up by land cover class and NLDAS model used. We used the RMSEs provided in ref. 53 for the VIC and Mosaic models and found the resultant error of the models averaged together by propagating the RMSEs ($\sigma$) across

each land cover type as:

$$\sigma_{NLDAS} = \frac{1}{2}\sqrt{\sigma_{VIC}^2 + \sigma_{Mosaic}^2} \qquad (3)$$

We determined an expected human ET error by propagating the RMSEs for OpenET and NLDAS ET as:

$$\sigma_{Human} = \sqrt{\sigma_{OpenET}^2 + \sigma_{NLDAS\ ET}^2} \qquad (4)$$

Some land cover classifications (e.g., barren land, developed land) did not have reported error metrics. In these cases, we used the reported mean RMSEs across all land cover types. All RMSE values and sources are provided in Table 1.

In most cases, the standard deviation of the data within different land cover classes was less than the anticipated error from the RMSE values previously reported and propagated in this study (see section "Results"). Therefore, we report the mean value alongside the reported standard deviation and offer expected error metrics as a useful point of comparison.

All fine-spatial resolution datasets (NLCD land cover data, OpenET, and VIIRS NDVI) were re-gridded to the coarse NLDAS $0.125 \times 0.125$-degree grid. For the land cover data, we used the most commonly occurring vegetation type data within a certain pixel using the 2023 NLCD land cover map. Although NLCD data are available every year, there was no change in land cover at the coarse spatial resolution of this study between the two years. For VIIRS NDVI and OpenET, we averaged all values that fell within a given NLDAS pixel. Although the regridding excludes some vegetation types based on coverage, these represent less than 3% of the land surface area. An additional concern with this approach is the aggregation of heterogeneous data within a coarse-resolution NLDAS pixel. To test this, we computed change statistics for NDVI with and without aggregation and found negligible (<1%) differences in our results, except for vegetation types with fewer than 10 pixels at the coarse resolution data. Therefore, for consistency, all results are reported at the NLDAS resolution and exclude all vegetation classes that do not have more than 10 pixels across the state (although these are still included in maps for visualization).

For all our analyses, we used the growing season average (May–September) to capture the peak of ET and vegetation productivity in the years of interest. Furthermore, monthly data tends to be more accurate than daily or instantaneous ET due to the cancellation of errors and reduced uncertainty in energy balance closure of validation data[52,104,105]. As a point of reference, we established a growing season average historical baseline by averaging the May–September monthly data from 2016 to 2021 for all datasets of interest. This baseline includes years with both significantly above (2016, 2017, 2019) and below (2018, 2020, and 2021) average precipitation and is the longest possible record where all datasets are available. All reported uncertainty intervals are calculated as the standard deviation of data within a certain grouping to capture the spread of the data.

## Data availability
All data used in this study are publicly available for download. OpenET data are available at https://etdata.org/. NLDAS data are available from the NASS GES DISC https://ldas.gsfc.nasa.gov/data. Annual NLCD is available across a variety of platforms https://www.usgs.gov/centers/eros/science/data-access. VIIRS NDVI data is available through the Land Processes DAAC https://www.earthdata.nasa.gov/data/catalog/lpcloud-vnp13a3-002.

## Code availability
All data and code used in this analysis are archived at: https://doi.org/10.5281/zenodo.15103325.

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

## Acknowledgements

Z.A.P. was supported by an appointment to the NASA Postdoctoral Program at the Jet Propulsion Laboratory. A portion of this research was carried out at the Jet Propulsion Laboratory, California Institute of Technology, under a contract with the National Aeronautics and Space Administration. © 2025 All rights reserved. Government sponsorship is acknowledged.

## Author contributions

Z.A.P. contributions include data preparation, software development, project conceptualization, project management, manuscript writing original draft, reviewing, and editing. R.G.'s contributions include data preparation, software development, project conceptualization, manuscript writing original draft, reviewing, and editing. A.B.'s contributions include manuscript review and editing. S.R.'s contributions include manuscript review and editing. C.L.'s contributions include project conceptualization and manuscript review, and editing. J.T.R.'s contributions include project conceptualization and manuscript review, and editing. K.C.N.'s contributions include manuscript review and editing.

## Competing interests

The authors declare no competing interests.
