## [Transparent Peer Review file · Communications Sustainability]

Human contributions to evapotranspiration mitigate swings in dry to wet year transitions

Corresponding Author: Dr Zoe Pierrat

This manuscript has been previously reviewed at another Nature Portfolio journal. This document only contains reviewer comments and rebuttal letters for versions considered at Communications Sustainability.

Version 0:

Decision Letter:

Dear Dr Pierrat,

Your manuscript titled "Human contributions to evapotranspiration mitigate swings in dry to wet year transitions" has now been seen by 2 reviewers, whose comments are appended below. You will see that they find your work of some potential interest. However, they have raised quite substantial concerns that must be addressed. In light of these comments, we cannot accept the manuscript for publication, but would be interested in considering a revised version that fully addresses these serious concerns. Specifically, a revised manuscript must:

1. Fully clarify California's land cover classification to enable accurate interpretation of evapotranspiration sources across natural and human-modified landscapes.
2. Fully justify or revise the method of calculating human ET over natural vegetation, ensuring that it reflects anthropogenic impacts and focuses on managed or modified land covers.
3. Explicitly attribute ET changes to irrigation by incorporating "green" and "blue" water concepts and compellingly demonstrate the broader significance and applicability of the findings to enhance their relevance for water resource management and climate adaptation.

We hope you will find the reviewers' comments useful as you decide how to proceed. Should additional work allow you to address these criticisms, we would be happy to look at a substantially revised manuscript. If you choose to take up this option, please either highlight all changes in the manuscript text file, or provide a list of the changes to the manuscript with your responses to the reviewers.

When resubmitting, please provide a point-by-point response to the reviewers' comments. Please submit your responses as a separate file, distinct from your cover letter where you can add responses to the Editors' comments that you do not want to be made available to the reviewers. Word files are preferred. We recommend that any figures, tables or graphs that are included in the response to reviewers are also included in the main article or Supplementary Information.

If the revision process takes significantly longer than three months, we will be happy to reconsider your paper at a later date, as long as nothing similar has been accepted for publication at Communications Sustainability or published elsewhere in the meantime.

Please use the following link to submit your revised manuscript, point-by-point response to the reviewers' comments with a

list of your changes to the manuscript text (which should be in a separate document to any cover letter), a tracked-changes version of the manuscript (as a PDF file) and any completed checklist:

Link Redacted

Please do not hesitate to contact us if you have any questions or would like to discuss the required revisions further. Thank you for the opportunity to review your work.

Best regards,

Alireza Bahadori, PhD
Consulting Editor
Communications Sustainability
Associate Editor
Communications Earth & Environment
@commssust.bsky.social

EDITORIAL POLICIES AND FORMAT

If you decide to resubmit your paper, please ensure that your manuscript complies with our editorial policies and complete and upload the checklist below as a Related Manuscript file type with the revised article:

Editorial Policy Policy requirements
(Download the link to your computer as a PDF.)

- Behavioural and social science
- Ecological, evolutionary & environmental sciences
- Life sciences

<https://www.nature.com/documents/nr-reporting-summary.zip>

For your information, you can find some guidance regarding format requirements summarized on the following checklist: (<https://www.nature.com/documents/commsj-phys-style-formatting-checklist-article.pdf>) and formatting guide (<https://www.nature.com/documents/commsj-phys-style-formatting-guide-accept.pdf>).

REVIEWER COMMENTS:

Reviewer #1 (Remarks to the Author):

The paper's current form does not adequately support its ambitious title, "Human Contributions to Evapotranspiration Mitigate Swings in Dry to Wet Year Transitions". Major revisions are required to address outdated literature, superficial treatment of human interventions, inconsistent methodological choices, and incomplete referencing. Without these substantial changes, the study's conclusions lack robustness, and its contribution to the field remains unconvincing. I urge the authors to undertake a thorough restructuring to align the analysis with the stated objectives and strengthen its scientific foundation.

Comments to the respected authors:

1- The literature review is reasonably comprehensive but would benefit from the inclusion of more recent studies, particularly those published in 2023 & 2024. For example, the discussion around line 100 cites studies only up to 2022, which risks overlooking pertinent advancements in the field from the past year. Incorporating these could strengthen the paper's relevance and contextual grounding.

2- The paper should explicitly address human interventions/influences in the following sections:

2.1- Consider revising the statement to: "Understanding how both natural and human contributions to the ET signal change in California because of changing environmental conditions will help us anticipate and manage future changes to the water budget."

Suggest rephrasing to: "Understanding how both natural and human contributions to the ET signal change in California because of changing environmental conditions and human interventions will help us anticipate and manage future changes

to the water budget."

2.2- Consider revising the statement to: " Therefore, understanding the ET response to climatic swings in California, which are expected to become more extreme and more frequent"

Suggest rephrasing to: " Therefore, understanding the ET response to climatic swings and human influences in California, which are expected to become more extreme and more frequent"

3- The study notes that California's land cover is dominated by shrub/scrub (42.7% of regrided pixels), evergreen forest (17.6%), grassland/herbaceous (14.9%), cultivated crops (9.9%), and developed, medium-intensity land (2.3%). This raises several points for clarification:

3.1- Does the study classify shrub/scrub, evergreen forest, and grassland/herbaceous as natural vegetation types unaffected by human interventions, while designating cultivated crops and medium-intensity developed land as human-influenced areas where ET reflects anthropogenic impacts? Please explicitly clarify the categorization and its implications for interpreting total ET contributions.

3.2- The estimation of human-based ET over natural vegetation raises questions about the study's methodological approach. Specifically, what is the rationale for defining "Human ET = OpenET - NLDAS ET" over natural vegetation types like evergreen forests? This approach seems counterintuitive, as human-induced ET effects would logically be more relevant to human-modified land covers (e.g., croplands) or managed forest systems (e.g., agroforestry). Please justify this choice or reconsider its application to ensure alignment with the study's focus on human contributions.

4- The reference list omits key citations, namely (Xia et al., 2012a; 2012b) and (Melton et al., 2022), which hinders a thorough evaluation of the study's methodological foundation and contextual support. Including these references is essential for transparency and to allow reviewers to assess the cited methods and conclusions concerning your work. Please update the reference list accordingly.

Reviewer #2 (Remarks to the Author):

Review for "Human contributions to evapotranspiration mitigate swings in dry to wet year transitions". This paper uses satellite resources to estimate California statewide ET in 2022 and 2023 growing season months, and partitions the ET into "natural" and "human" ET. The goal of the paper seems to be to quantify human contribution to ET, and how this contribution differs in dry vs. wet years.

The topic is interesting and relevant to the journal. The manuscript is generally well-written, although there are some general concerns regarding the terminology that is used. Namely, use of "human" ET so simply replace the word irrigation, is problematic. Yes, irrigation is a human-influenced management practice that increases ET, but so is agriculture in general. I suggest the manuscript be revised to specify the influence of irrigation, rather than generally stating "human" influence. My guess is that the authors chose this title and strategy to imply broader focus, but the paper would be more focused if you simply call it what it is. Also, there are already terms used to describe the impact of ET from different sources, called "green" and "blue" water, which would be good to cite and would be more appropriate to use than "human" ET.

The other issue I have with the paper is that the outcome is quite intuitive - in a dry year, more irrigation is needed, therefore the "human" (i.e., irrigation) influence on ET is increased. While the application of ET models in this context is certainly interesting to see, what is the impact of the outcome? There is some discussion that management changes under water limitations can mitigate high ET, but is that it? The paper would be served well if the authors could make a more convincing argument as to the importance of the outcomes, and how/why the process could/should be repeated in other contexts.

I have made specific comments and suggested edits in a separate Word file.

Communications Sustainability is committed to improving transparency in authorship. As part of our efforts in this direction, we are now requesting that all authors identified as 'corresponding author' create and link their Open Researcher and Contributor Identifier (ORCID) with their account on the Manuscript Tracking System prior to acceptance. ORCID helps the scientific community achieve unambiguous attribution of all scholarly contributions. You can create and link your ORCID from the home page of the Manuscript Tracking System by clicking on 'Modify my Springer Nature account' and following the instructions in the link below. Please also inform all co-authors that they can add their ORCIDs to their accounts and that they must do so prior to acceptance.

Version 1:

Decision Letter:

Dear Dr Pierrat,

Your revised manuscript titled "Human contributions to evapotranspiration mitigate swings in dry to wet year transitions" has now been seen by our original reviewer 1 and a new reviewer 3 who replaces the original reviewer 2 (reviewer 2 was not available for further comments). All comments appear below. You will see that the reviewers appreciate the effort you put in the revisions, but reviewer 3 raises important concerns, particularly regarding the ET partitioning approach used. We are interested in the possibility of publishing your study in Communications Sustainability, but would like to consider your responses to these concerns and assess a revised manuscript before we make a final decision on publication.

We therefore invite you to revise and resubmit your manuscript, along with a point-by-point response that takes into account the points raised. Please highlight all changes in the manuscript text file.

Please submit your point-by-point responses as a separate file, distinct from your cover letter where you can add responses to the Editors' comments that you do not want to be made available to the reviewers. Word files are preferred. We recommend that any figures, tables or graphs that are included in the response to reviewers are also included in the main article or Supplementary Information.

Please use the following link to submit your revised manuscript, point-by-point response to the referees' comments (which should be in a separate document to any cover letter), a tracked-changes version of the manuscript (as a PDF file) and the completed checklist:

Link Redacted

We hope to receive your revised paper within six weeks; please let us know if you aren't able to submit it within this time so that we can discuss how best to proceed. If we don't hear from you, and the revision process takes significantly longer, we may close your file. In this event, we will still be happy to reconsider your paper at a later date, as long as nothing similar has been accepted for publication at Communications Sustainability or published elsewhere in the meantime.

Please do not hesitate to contact us if you have any questions or would like to discuss these revisions further. We look forward to seeing the revised manuscript and thank you for the opportunity to review your work.

Best regards,

Alireza Bahadori, PhD
Consulting Editor
Communications Sustainability
Associate Editor
Communications Earth & Environment
@commssust.bsky.social

EDITORIAL POLICIES AND FORMATTING

- Behavioural and social science
- Ecological, evolutionary & environmental sciences
- Life sciences

Furthermore, please align your manuscript with our format requirements, which are summarized on the following checklist: <https://www.nature.com/documents/commsj-phys-style-formatting-checklist-article.pdf> Communications Sustainability formatting checklist

and also in our style and formatting guide <https://www.nature.com/documents/commsj-phys-style-formatting-guide-accept.pdf> Communications Sustainability formatting guide .

*** DATA: Communications Sustainability endorses the principles of the Enabling FAIR data project (<http://www.copdess.org/enabling-fair-data-project/>). We ask authors to make the data that support their conclusions available in permanent, publicly accessible data repositories. (Please contact the editor if you are unable to make your data available).

All Communications Sustainability manuscripts must include a section titled "Data Availability" at the end of the Methods section or main text (if no Methods). More information on this policy, is available at <http://www.nature.com/authors/policies/data/data-availability-statements-data-citations.pdf>.

If a community resource is unavailable, data can be submitted to generalist repositories such as <https://figshare.com/> or <http://datadryad.org/> Dryad Digital Repository. Please provide a unique identifier for the data (for example a DOI or a permanent URL) in the data availability statement, if possible. If the repository does not provide identifiers, we encourage authors to supply the search terms that will return the data. For data that have been obtained from publicly available sources, please provide a URL and the specific data product name in the data availability statement. Data with a DOI should be further cited in the methods reference section.

REVIEWER COMMENTS:

Reviewer #1 (Remarks to the Author):

I would like to express my gratitude to the authors for their thorough and constructive response to the feedback provided. Their revisions have effectively addressed the concerns raised, resulting in a significantly improved manuscript.

After carefully reviewing the current version, I am satisfied with the clarity, rigor, and overall quality of the work. Accordingly, I recommend the manuscript for publication in its present form, as it makes a valuable contribution to the field.

Thank you for the opportunity to review this work.

Sincerely,

Reviewer #3 (Remarks to the Author):

This is a revised manuscript exploring an interesting topic. While the authors have addressed prior critiques with substantial modifications, fundamental concerns remain:

First, for managed ecosystems (particularly irrigated agriculture), sustaining crop productivity requires substantial water inputs—irrespective of green/blue water sources. Consequently, ET, representing almost agricultural water consumption, must be maintained near optimal levels to preserve economic yields. This intrinsic requirement explains the documented irrigation surge during droughts.

The study fails to quantify how anthropogenic water use (especially irrigation) responds to hydroclimate variability (e.g., dry to wet transitions) in water-scarce regions like California. More crucially, it neglects to assess how the consumed blue water will affect the local water balance as well as the ecosystem. The authors assessed human ET but ignoring its downstream impact.

Second, methodological ambiguity exists in ET Partitioning and the current assumption may not work. The derivation of

"human ET" as the arithmetic difference between OpenET and NLDAS products lacks robust justification. 1) The algorithms to obtain the two ET products were unclear, preventing our understanding why human ET equals to their difference. 2) No evidence confirms NLDAS represents purely natural ET in human-modified landscapes. 3) Performance of both ET products were unclear, particularly in irrigation hotspots such as the central valley.

There are some specific comments listing as follows:

The numbers in front of the comments indicate line number.

1. L60. Incorrect citation "P W Liu" and similar ones across the manuscript.
2. L69. ET is important. How about the basic water components in California. Such as annual rainfall, irrigated area, and how about agriculture water consumption. Is any study showing the relationship between ET and groundwater extraction?
3. L148. Inconsistency between NLDAS and NDLAS.
4. L217. What does FHI represent? Please consider defining the abbreviation at the first time.
5. L348. Is any text showing the change of crops?

Communications Sustainability is committed to improving transparency in authorship. As part of our efforts in this direction, we are now requesting that all authors identified as 'corresponding author' create and link their Open Researcher and Contributor Identifier (ORCID) with their account on the Manuscript Tracking System prior to acceptance. ORCID helps the scientific community achieve unambiguous attribution of all scholarly contributions. You can create and link your ORCID from the home page of the Manuscript Tracking System by clicking on 'Modify my Springer Nature account' and following the instructions in the link below. Please also inform all co-authors that they can add their ORCIDs to their accounts and that they must do so prior to acceptance.

If you experience problems in linking your ORCID, please contact the Platform Support Helpdesk.

Version 2:

Decision Letter:

Dear Dr Pierrat,

Your revised manuscript titled "Human contributions to evapotranspiration mitigate swings in dry to wet year transitions" has now been seen by our reviewers, whose comments appear below. In light of their advice we are delighted to say that we are happy, in principle, to publish a suitably revised version in Communications Sustainability, provided you acknowledge the limitation of NLDAS in explicit representation of irrigation and the uncertainty this introduces in evapotranspiration estimates, and tone down conclusions accordingly, along the line recommended by our reviewer 3.

If you can address this request, we therefore invite you to revise your paper to comply with our format requirements and to maximise the accessibility and therefore the impact of your work.

EDITORIAL REQUESTS:

****Please take care to match our formatting and policy requirements. We will check revised manuscript and return manuscripts that do not comply. Such requests will lead to delays. ****

SUBMISSION INFORMATION:

In order to accept your paper, we require the files listed at the end of the Editorial Requests Table; the list of required files is also available at <https://www.nature.com/documents/commsj-file-checklist.pdf> .

OPEN ACCESS:

Communications Sustainability is a fully open access journal. Articles are made freely accessible on publication. For further information about article processing charges, open access funding, and advice and support from Nature Portfolio, please visit <https://www.nature.com/commssustain/open-access>

Link Redacted

Best regards,

Alireza Bahadori, PhD
Consulting Editor
Communications Sustainability
Associate Editor
Communications Earth & Environment
@CommsSustain.nature.com

REVIEWERS' COMMENTS:

Reviewer #3 (Remarks to the Author):

The authors have revised the manuscript. However, I remain concerned about the methodology for partitioning ET, specifically the derivation of "human ET" as the arithmetic difference between OpenET and NLDAS ET products. The authors stated that "NLDAS does not simulate irrigation" and that this "lack of explicit irrigation representation is a key source of model error". This only confirms NLDAS's unsuitability for irrigated lands; it does not imply that ET over irrigated land represents solely natural water loss. Moreover, the algorithms adopted for estimating OpenET also do not explicitly simulate irrigation. While remote sensing data (used in OpenET) captured at an instant time might reflect irrigation effects, upscaling instantaneous ET to daily/monthly scales may not fully account for irrigation occurring later in the period, introducing uncertainty into monthly ET estimates. Although OpenET validation shows an R^2 of 0.9, the average normalized MAE (RMSE) was 17% (22%). Validation at orchard sites in the Central Valley, California, indicates similar uncertainty levels.

The final decision remains with the editor. Nonetheless, the authors should explicitly acknowledge the limitations outlined above.

** Visit Nature Portfolio's author and referees' website at <http://www.nature.com/authors> for information about policies, services and author benefits**

Response to reviewers for “Human contributions to evapotranspiration mitigate swings in dry to wet year transitions”

We thank the editor and reviewers for their careful consideration of our manuscript. We have reviewed the feedback and offer a significantly revised version that addresses the concerns brought up during the review process. Key changes include:

- 1. A fully revised introduction which incorporates blue and green water concepts, clarifies the various ways humans can impact ET beyond irrigation, and provides a brief description of our methodological approach with relevant references.*
- 2. An evaluation of errors and uncertainties in our results based on expected RMSEs. Expected RMSEs come from previous work and are propagated through our analysis. These results explain concerns over calculating human ET and fractional human impact over more “natural” vegetation and negative values for human ET.*
- 3. Improvements to the overall presentation of the study including more updated/recent citations, further details in the methods, one additional figure emphasizing the climatic swings from 2022-2023, and re-writes for clarity throughout the text.*

We believe these changes have improved the manuscript and addressed key concerns brought forth by the reviewers. More details on these changes, including specific responses to editorial and reviewer concerns are found below italicized. We look forward to your response and re-consideration of our manuscript.

EDITOR COMMENTS:

Dear Dr Pierrat,

Your manuscript titled "Human contributions to evapotranspiration mitigate swings in dry to wet year transitions" has now been seen by 2 reviewers, whose comments are appended below. You will see that they find your work of some potential interest. However, they have raised quite substantial concerns that must be addressed. In light of these comments, we cannot accept the manuscript for publication, but would be interested in considering a revised version that fully addresses these serious concerns. Specifically, a revised manuscript must:

1. Fully clarify California’s land cover classification to enable accurate interpretation of evapotranspiration sources across natural and human-modified landscapes.

We have clarified the spatial aggregation of NLDAS land cover classifications in lines 1014-1021:

“An additional concern with this approach is the aggregation of heterogeneous data within a coarse resolution NLDAS pixel. To test this, we computed change statistics for NDVI with and without aggregation and found negligible (<1%) differences in our results, except for vegetation types with fewer than 10 pixels at

the coarse resolution data. Therefore, for consistency, all results are reported at the NLDAS resolution and exclude all vegetation classes that do not have more than 10 pixels across the state (although these are still included in maps for visualization).”

Additionally, we have added a new section where we quantify uncertainty in our reported values against the anticipated uncertainty as a result of previously reported RMSE. We find that in all cases the standard deviation observed in our results is less than the expected RMSE based on previous work. This is detailed in our methods section, and we have added additional text to the results in lines 372-503 explaining this finding:

“Notably, the observed standard deviation for each land cover classification is less than the expected RMSE based on previous work quantifying uncertainty in these datasets (Volk et al., 2024; Xia et al., 2015; B. Zhang et al., 2020) and propagated through our analysis (Table 1). The lower standard deviations give us confidence that additional error introduced by the aggregation of multiple land cover classes within a single pixel is insignificant compared with the inherent uncertainty in the data themselves. Furthermore, for land cover classes that we would not expect to have a human influence on ET (shrub/scrub, barren land, grassland/herbaceous), human ET is not statistically different from zero (i.e., human ET=0 is within the average \pm the standard deviation of reported values). The one exception to this is in evergreen needleleaf forests which could have some human influence because of forest thinning practices or spatial aggregation across cropland and developed land. However, the propagated uncertainty based on the OpenET and NLDAS datasets in evergreen needleleaf forests is 30 mm month⁻¹, suggesting that human ET in evergreen forests is biased (more on this in Section 3), but not statistically different from zero.”

Lastly, we have added clarification on why we calculate human ET over “natural” vegetation as an approach for accounting for human impacts on ET within the coarse NLDAS pixel that might go unaccounted for otherwise in lines 954-960:

“By calculating human ET over land cover classes that are not expected to have a human influence on ET (e.g., evergreen needleleaf forests), we are able to 1) provide a statistical baseline reference for our results (i.e., do natural landcover types show human ET = 0); and 2) account for human ET in agricultural and developed land that might go unaccounted for in the spatial aggregation of heterogeneous land cover classes within the coarse NLDAS pixel (Section 4.2).”

2. Fully justify or revise the method of calculating human ET over natural vegetation, ensuring that it reflects anthropogenic impacts and focuses on managed or modified land covers.

We thank the reviewers and editor for noting that it is counterintuitive to calculate human ET over natural landcover, however, we believe this allows for several significant advantages:

1. *This framework builds on previous studies. Computing human and non-human contributions to ET (or “green” and “blue” water contributions) over entire landscapes has been demonstrated and proven useful in several studies. This is in part because it allows for the detection of irrigation or human impacts beyond what is reported by local or state data (see points 2 and 3).*
 - . Velpuri, N. M., & Senay, G. B. (2017). Partitioning evapotranspiration into green and blue water sources in the conterminous United States. *Scientific Reports*, 7. <https://doi.org/10.1038/s41598-017-06359-w>
 - . Pascolini-Campbell, M., Fisher, J. B., & Reager, J. T. (2021). GRACE-FO and ECOSTRESS Synergies Constrain Fine-Scale Impacts on the Water Balance. *Geophysical Research Letters*, 48(15), e2021GL093984. <https://doi.org/10.1029/2021GL093984>
 - . Boser, A., Caylor, K., Larsen, A., Pascolini-Campbell, M., Reager, J. T., & Carleton, T. (2024). Field-scale crop water consumption estimates reveal potential water savings in California agriculture. *Nature Communications*, 15(1), 2366. <https://doi.org/10.1038/s41467-024-46031-2>
2. *Our method does not make assumptions about or rely on additional data on soil, crop characteristics, national statistics, reports, crop-related maps, and actual irrigation as input variables. This enables us to evaluate the variability in the models beyond agricultural regions and quantify both the statistical error in the models as well as typically managed regions.*
3. *This approach allows us to quantify fine-scale impacts on the water balance coming from smaller agricultural regions which may fall into an NDLAS pixel but not be the dominant vegetation type. Humans have dramatically impacted the majority of Californian landscapes. Our approach enables us to identify regions where human activities may impact the water balance even if the vegetation type is not one typically thought of as being modified by humans significantly.*

We have fully revised our introduction, discussion, and methods sections to reflect these points with specific examples noted below in the responses to reviewer comments.

3. Explicitly attribute ET changes to irrigation by incorporating “green” and “blue” water concepts and compellingly demonstrate the broader significance and applicability of the findings to enhance their relevance for water resource management and climate adaptation.

We thank the reviewers for referencing the terms “green” and “blue” water and agree that these are pertinent topics to include in the manuscript. We have included these concepts now in our introduction and discussion and feel that it (1) acknowledges the complexity of understanding anthropogenic / environmental controls on ET and (2) provides an appropriate segue for future work and underscores this as a caveat in our current approach. We now introduce the blue /

green water concepts in lines 144-161 and expand on these ideas further in the discussion:

“ET is controlled by environmental factors including temperature, soil moisture, solar radiation, wind, and atmospheric vapor pressure (Bento et al., 2018; Fisher et al., 2008; Joiner et al., 2018); biotic factors like ecosystem type, vegetation health, and plant functional traits (Bhattarai & Wagle, 2021; Brown et al., 2010; Detto et al., 2006); and water applied via irrigation (Boser et al., 2024; Pascolini-Campbell et al., 2021). Accordingly, ET is a function of both natural (environmental controls and vegetation health) and human (management decisions such as forest thinning or crop selection, and irrigation practices) contributions. Similarly, the water source used to support ET can come from both natural or human sources – termed “green” and “blue” water. Green water is precipitation that does not run off and temporarily contributes to soil water storage. Blue water is surface and groundwater stored in rivers, lakes, aquifers and dams, and can be extracted for human use (Falkenmark & Rockström, 2006; Mao et al., 2020). The relative contributions of green and blue water to ET are therefore important for anticipating and managing future changes to the water budget and future water resources. Importantly, both the amount of water available to support ET from both green and blue water sources, as well as the natural and environmental controls on ET are all sensitive to the rapidly changing environmental conditions in California.”

Because it was not in our study scope to explore blue / green, we maintained “human” and “natural” terminology for several reasons:

- 1. The blue and green water concepts describe the source of the water used. However, our method cannot distinguish the source of the water, only whether it is accounted for in a natural/physics-based model. Therefore, it would not be accurate to revise the manuscript throughout to correspond with blue / green concepts.*
- 2. Our methodology is directly adapted from and thus consistent with prior research (Zeng et al 2022, Zou et al 2017, Pascolini et al 2021, Chen et al 2017, Boser et al 2022) which distinguishes between “human” and “natural” components of ET, and also do not include sources of water or corresponding terminology (“blue”/“green”)*
 - . Zeng, H., Elnashar, A., Wu, B., Zhang, M., Zhu, W., Tian, F., & Ma, Z. (2022). A framework for separating natural and anthropogenic contributions to evapotranspiration of human-managed land covers in watersheds based on machine learning. *Science of The Total Environment*, 823, 153726. <https://doi.org/10.1016/j.scitotenv.2022.153726>*
 - . Zou, M., Niu, J., Kang, S., Li, X., & Lu, H. (2017). The contribution of human agricultural activities to increasing evapotranspiration is significantly greater than climate change effect over Heihe agricultural region. *Scientific Reports*, 7(1), 8805. <https://doi.org/10.1038/s41598-017-08952-5>*

- Chen, X., Mo, X., Hu, S., & Liu, S. (2017). Contributions of climate change and human activities to ET and GPP trends over North China Plain from 2000 to 2014. *Journal of Geographical Sciences*, 27(6), 661–680. <https://doi.org/10.1007/s11442-017-1399-z>
- Pascolini-Campbell, M., Fisher, J. B., & Reager, J. T. (2021). GRACE-FO and ECOSTRESS Synergies Constrain Fine-Scale Impacts on the Water Balance. *Geophysical Research Letters*, 48(15), e2021GL093984. <https://doi.org/10.1029/2021GL093984>
- Boser, A., Caylor, K., Larsen, A., Pascolini-Campbell, M., Reager, J. T., & Carleton, T. (2024). Field-scale crop water consumption estimates reveal potential water savings in California agriculture. *Nature Communications*, 15(1), 2366. <https://doi.org/10.1038/s41467-024-46031-2>

REVIEWER COMMENTS:

Reviewer #1 (Remarks to the Author):

The paper's current form does not adequately support its ambitious title, "Human Contributions to Evapotranspiration Mitigate Swings in Dry to Wet Year Transitions". Major revisions are required to address outdated literature, superficial treatment of human interventions, inconsistent methodological choices, and incomplete referencing. Without these substantial changes, the study's conclusions lack robustness, and its contribution to the field remains unconvincing. I urge the authors to undertake a thorough restructuring to align the analysis with the stated objectives and strengthen its scientific foundation.

Comments to the respected authors:

1- The literature review is reasonably comprehensive but would benefit from the inclusion of more recent studies, particularly those published in 2023 & 2024. For example, the discussion around line 100 cites studies only up to 2022, which risks overlooking pertinent advancements in the field from the past year. Incorporating these could strengthen the paper's relevance and contextual grounding.

We have extensively revised our manuscript (most notably in the introduction and discussion) to include a significant number of more recent publications (>10 from 2023 or later) and bring the contextual grounding of our paper up to speed with the most up-to-date literature. Should there be any additional articles to consider, we would be appreciative of any further recommendations from the reviewer.

2- The paper should explicitly address human interventions/influences in the following sections:

We note that both reviewers wanted clarification on human interventions/influences on ET. Therefore, we have significantly revised the introduction to more clearly explain the variety of ways humans can impact the

ET signal. Most notably, we have included a more complete description of ET's role in the total water budget (lines 130-143), explained its relationship with blue and green water (lines 141-161), and expanded on our definition of "human" ET to more explicitly include management decisions such as forest thinning or crop selection (lines 149-151, 184-270, 953-960).

2.1- Consider revising the statement to: "Understanding how both natural and human contributions to the ET signal change in California because of changing environmental conditions will help us anticipate and manage future changes to the water budget." Suggest rephrasing to: "Understanding how both natural and human contributions to the ET signal change in California because of changing environmental conditions and human interventions will help us anticipate and manage future changes to the water budget."

Because we have extensively revised our manuscript, this statement no longer appears in the revised paper. Instead, we have a more complete description of human interventions and influences in both the introduction and discussion.

2.2- Consider revising the statement to: " Therefore, understanding the ET response to climatic swings in California, which are expected to become more extreme and more frequent"

Suggest rephrasing to: " Therefore, understanding the ET response to climatic swings and human influences in California, which are expected to become more extreme and more frequent"

The suggested phrasing from the reviewer would suggest that "human influences in California" would become more extreme and more frequent which is not the intent of this statement. However, we agree that we can emphasize human intervention as a factor that will respond to these climatic swings. Therefore, we have revised several portions of this paragraph to emphasize human intervention as a response to climatic swings including a more complete description of irrigation changes (lines 171-179) and potential changes in ET as a result of forest thinning under drought (lines 180-270).

3- The study notes that California's land cover is dominated by shrub/scrub (42.7% of regrided pixels), evergreen forest (17.6%), grassland/herbaceous (14.9%), cultivated crops (9.9%), and developed, medium-intensity land (2.3%). This raises several points for clarification:

3.1- Does the study classify shrub/scrub, evergreen forest, and grassland/herbaceous as natural vegetation types unaffected by human interventions, while designating cultivated crops and medium-intensity developed land as human-influenced areas where ET reflects anthropogenic impacts? Please explicitly clarify the categorization and its implications for interpreting total ET contributions.

We have clarified that this study explores natural and human impacts on ET across the entire state of California, regardless of vegetation type. This is because a large portion of California land can be considered "working" landscapes and very little of the land is entirely unaffected by human impact. As

of 2022 only 24% of California land was designated conserved area
https://resources.ca.gov/-/media/CNRA-Website/Files/Initiatives/30-by-30/Final_Pathwaysto30x30_042022_508.pdf.

Two key areas we have modified (although this is a non-exhaustive list) are:

In the introduction in lines 308-312:

“This approach is applied across the entire state of California regardless of vegetation class to assess both the uncertainty in datasets used (Volk et al., 2024; Xia et al., 2015; B. Zhang et al., 2020) and potential human impacts on ET in California’s working landscapes beyond the agricultural sector (Velpuri & Senay, 2017).”

And in the results in lines 378-503:

“Furthermore, for land cover classes that we would not expect to have a human influence on ET (shrub/scrub, barren land, grassland/herbaceous), human ET is not statistically different from zero (i.e., human ET=0 is within the average \pm the standard deviation of reported values). The one exception to this is in evergreen needleleaf forests which could have some human influence because of forest thinning practices or spatial aggregation across cropland and developed land. However, the propagated uncertainty based on the OpenET and NLDAS datasets in evergreen needleleaf forests is 30 mm month⁻¹, suggesting that human ET in evergreen forests is biased (more on this in Section 3), but not statistically different from zero.”

3.2- The estimation of human-based ET over natural vegetation raises questions about the study’s methodological approach. Specifically, what is the rationale for defining "Human ET = OpenET - NLDAS ET" over natural vegetation types like evergreen forests? This approach seems counterintuitive, as human-induced ET effects would logically be more relevant to human-modified land covers (e.g., croplands) or managed forest systems (e.g., agroforestry). Please justify this choice or reconsider its application to ensure alignment with the study’s focus on human contributions.

The rationale for quantifying Human ET as the difference between OpenET and NLDAS ET is that it allows us to assess both the uncertainty in datasets used and potential human impacts on ET in California’s working landscapes beyond the agricultural sector. We have extensively modified the introduction and discussion of the paper to more clearly explain how humans may be modifying ET in ways that are not directly related to irrigation. For example, forest thinning during drought has been proposed as a way to aid in forest resiliency although the subsequent impacts of this on the water budget are highly uncertain. One example of this is in lines 180-270.

“Non-agricultural regions may also experience increased ET as a result of higher evaporative demand (Zhao et al., 2022). On the other hand, ET can also be

reduced under extreme drought, driven by a decline in available water sources for soil evaporation and plant transpiration, as well as vegetation mortality (He et al., 2022; Seneviratne et al., 2010). In forests, forest thinning has been proposed as an effective drought stress management decision, as it has been shown to increase forest resiliency under drought (Sankey & Tatum, 2022). The subsequent impacts of thinning on ET show varied results (del Campo et al., 2022). Prior work has reported both a net decrease in ET as a result of the reduction in tree cover (Roche et al., 2018); and no change/increased in ET as a result of higher individual tree transpiration compensating for the loss in tree area, and higher wind speed and solar radiation on the forest floor (del Campo et al., 2019; X. Liu et al., 2018; Simonin et al., 2007). Enhanced precipitation and total water storage anomalies following drought can also improve vegetation recovery and rebalancing of ET (Au et al., 2023; Zhao et al., 2022). A vegetation boom following high precipitation will increase ET but may also contribute to an enhanced risk of wildfire, further altering the water budget (Barnard et al., 2023; Ma et al., 2020; Swain, 2021).”

Furthermore, because we are constrained to the coarse resolution of NLDAS there are many instances where a pixel classified as “evergreen forest includes agricultural land or agroforestry land when highly heterogenous landscapes are aggregated into one pixel. We do not wish to exclude these regions from our analysis. By retaining landcover classes which are not as expected to be impacted by humans (such as grasslands), we can retain the information about how much humans are impacting the water budget even over heterogenous regions within the NDLAS pixel that include both managed and non-managed landscapes (including agricultural areas and agroforestry). We have included discussion on these points in lines 775-782:

“In shrub/scrub, barren land, grassland/herbaceous, evergreen forests, humans did not have a statistically significant impact on the water budget. This suggests that any management decisions that may impact ET (such as forest thinning) are currently inconsequential to the state water budget, and natural ecosystems are primarily responding to natural controls. Although management decisions did not result in changes in ET at the spatial scale of this study, these decisions may still have an impact on local water budgets by modifying green water stored in the soil and moved through plants.”

And lines 868-870:

“However, the spread in the data was less than the expected RMSE based on previous work, suggesting that the spatial aggregation is less important than improving the data products themselves.”

4- The reference list omits key citations, namely (Xia et al., 2012a; 2012b) and (Melton et al., 2022), which hinders a thorough evaluation of the study’s methodological foundation and contextual support. Including these references is essential for

transparency and to allow reviewers to assess the cited methods and conclusions concerning your work. Please update the reference list accordingly.

We apologize for the omission of key citations and completely agree with the reviewer that a complete reference list is essential for transparency. The omission was a result of transferring the document from Google docs to Word for final formatting. We have resolved this issue and updated our citations list.

Reviewer #2 (Remarks to the Author):

Review for "Human contributions to evapotranspiration mitigate swings in dry to wet year transitions". This paper uses satellite resources to estimate California statewide ET in 2022 and 2023 growing season months, and partitions the ET into "natural" and "human" ET. The goal of the paper seems to be to quantify human contribution to ET, and how this contribution differs in dry vs. wet years.

The topic is interesting and relevant to the journal. The manuscript is generally well-written, although there are some general concerns regarding the terminology that is used. Namely, used of "human" ET so simply replace the word irrigation, is problematic. Yes, irrigation is a human-influenced management practice that increases ET, but so is agriculture in general. I suggest the manuscript be revised to specify the influence of irrigation, rather than generally stating "human" influence. My guess is that the authors chose this title and strategy to imply broader focus, but the paper would be more focused if you simply call it what it is. Also, there are already terms used to describe the impact of ET from different sources, called "green" and "blue" water, which would be good to cite and would be more appropriate to use than "human" ET.

*We are glad that the reviewer thinks our paper is interesting and well written. Regarding the terminology, we have specifically chosen **not** to use "irrigation" as we cannot directly attribute the observed changes in ET to irrigation compared with other potential human changes to ET (e.g., agricultural crop selection and other management practices as the reviewer suggests). Furthermore, we are calculating "human" ET over regions where we do not expect irrigation to have any impact (e.g., evergreen forests), but which might experience changes in ET because of other management decisions or processes not incorporated in the NLDAS model.*

We believe this misunderstanding comes from the way our methodological approach is described and the contextual framework for human ET is introduced in our introduction. To address this, we have almost entirely re-written our introduction to more specifically describe potential human contributions to the ET signal (beyond irrigation) and improved our methods to more appropriately explain how the human ET contribution is calculated (as the residual between total observed and modeled through natural processes). Additionally, we include how these terms relate to "green" and "blue" water sources.

To the reviewers point about “green” and “blue” water sources, generally, the “blue” and “green” water terms refer to the source of the water. An increased need for irrigation would indicate pressure on “blue” water resources (driven by a decrease in available “green” water sources). Oftentimes, the additional term “gray” water is also included to indicate water which has been reclaimed from contaminated sources. Our ET observations cannot distinguish the original source of the water, which could include blue or gray water sources as well as green water. Therefore, quantifying ET in these terms would be an inaccurate description.

The other issue I have with the paper is that the outcome is quite intuitive - in a dry year, more irrigation is needed, therefore the "human" (i.e., irrigation) influence on ET is increased. While the application of ET models in this context is certainly interesting to see, what is the impact of the outcome? There is some discussion that management changes under water limitations can mitigate high ET, but is that it? The paper would be served well if the authors could make a more convincing argument as to the importance of the outcomes, and how/why the process could/should be repeated in other contexts.

We have made extensive revisions to our manuscript to better highlight the impact of this study and include more recent references. Of note, the methodological approach used in this study has significant potential to help understand the water budget in other regions under a changing climate. One example of modified text to clarify the significance of this study is in lines 875-881:

“Our approach demonstrates the ability to capture changing contributions to ET from human and natural processes as a consequence of extreme climatic swings. As irrigation expands into water-stressed regions globally (McDermid et al., 2023; Mehta et al., 2024), understanding water resources is increasingly important. Delineating human and natural controls on ET will help us understand how natural climate controls will impact future water availability and whether human management decisions will be able to sustainably compensate for those changes.”

I have made specific comments and suggested edits in a separate Word file.

We have addressed all comments and suggested edits in the attached word file in a new track changes version of the document.

Response to reviewers for “Human contributions to evapotranspiration mitigate swings in dry to wet year transitions”

We thank the editor and reviewers for their careful consideration of our manuscript. We have reviewed the feedback and offer a revised version that addresses the concerns brought up during the review process.

We believe these changes have improved the manuscript and addressed concerns brought forth by reviewer 3. More details on these changes, including specific responses to reviewer concerns are found below italicized. We look forward to your re-consideration of our manuscript.

REVIEWER COMMENTS:

Reviewer #1 (Remarks to the Author):

I would like to express my gratitude to the authors for their thorough and constructive response to the feedback provided. Their revisions have effectively addressed the concerns raised, resulting in a significantly improved manuscript.

After carefully reviewing the current version, I am satisfied with the clarity, rigor, and overall quality of the work. Accordingly, I recommend the manuscript for publication in its present form, as it makes a valuable contribution to the field.

Thank you for the opportunity to review this work.

Sincerely,

We thank the reviewer for their positive view of our manuscript and the re-review of our publication. We also feel the reviewer’s constructive comments resulted in a significantly improved manuscript.

Reviewer #3 (Remarks to the Author):

This is a revised manuscript exploring an interesting topic. While the authors have addressed prior critiques with substantial modifications, fundamental concerns remain: First, for managed ecosystems (particularly irrigated agriculture), sustaining crop productivity requires substantial water inputs—irrespective of green/blue water sources. Consequently, ET, representing almost agricultural water consumption, must be maintained near optimal levels to preserve economic yields. This intrinsic requirement explains the documented irrigation surge during droughts.

The study fails to quantify how anthropogenic water use (especially irrigation) responds to hydroclimate variability (e.g., dry to wet transitions) in water-scarce regions like California.

We agree with the reviewer that for agricultural yields to remain stable during drought, agricultural water consumption (irrigation) must increase. This impact is demonstrated in our results (Figure 3 g&h). In this study, we quantify natural and human water use using the methods described in lines 428-491.

Anthropogenic ET's response to climate variability is quantified in our results in Figures 3 and 4, lines 247:

"Between 2022 and 2023, human ET decreased by an average of -14 ± 14 mm month⁻¹ (Figure 3i)."

And lines 270-275:

"Human ET shows a smaller change between years for land cover types which are not expected to have a human impact (barren land, shrub/scrub, grasslands, evergreen forests) while developed land and cultivated crops show a clear decrease in human ET in 2023 (44 ± 12 mm month⁻¹ in 2022 to 25 ± 16 mm month⁻¹ in 2023 for developed land; and 58 ± 19 mm month⁻¹ in 2022 to 36 ± 21 mm month⁻¹ in 2023 for cultivated crops) (Figure 4c)."

More crucially, it neglects to assess how the consumed blue water will affect the local water balance as well as the ecosystem. The authors assessed human ET but ignoring its downstream impact.

We agree with the reviewer that the downstream impacts of how changes to consumed blue and green water will affect the local water balance is a very interesting topic with many interconnected social and economic implications. We have added an additional discussion to this end which uses the results of this study as a foundation to guide future research in this area in lines 405-420:

"Delineating human and natural controls on ET will be increasingly important for managing blue and green water sources now and into the future. As irrigation expands into water-stressed regions globally^{93,94}, and water-stress is expected to increase⁹⁵, the downstream consequences of these changes to the entire water balance is largely unknown. By quantifying the human contribution to ET, we can better understand how human management decisions impact the entire water budget and the interplay between humans and natural climate influence. Prior work has demonstrated a decrease in groundwater storage during megadroughts⁷, but direct attribution of this depletion to human or natural controls can only be made by quantifying the human component of ET. Our approach demonstrated here can better link components and drivers of the water budget, including groundwater extraction and recharge, paving the way for understanding future water availability and guiding water resource management."

Second, methodological ambiguity exists in ET Partitioning and the current assumption may not work. The derivation of "human ET" as the arithmetic difference between OpenET and NLDAS products lacks robust justification.

We have modified our methods to more clearly justify our methodological approach with specific points to the reviewer's three main concerns below.

1) The algorithms to obtain the two ET products were unclear, preventing our understanding why human ET equals to their difference.

Detailed explanations on the algorithms producing ET and validation of these algorithms have been published in previous research (see Melton et al., 2022; Volk et al., 2023; 2024; Xia et al., 2014; 2015).

To make our methodological justification clearer, we have expanded the methods section to provide key details of the methodological approaches as relevant to this study. We provide relevant references to papers that provide more detailed explanations of model parameterizations and validation approaches for readers interested in more specifics.

For OpenET, we have revised lines 436-452:

“OpenET uses satellite-based land surface temperature and meteorological data from Landsat, Terra/Aqua MODIS, Suomi NPP VIIRS, GOES, and Sentinel-2^{49,52} to drive 6 different ET models. The models included in OpenET include both energy balance and reflectance-based approaches to ET mapping⁴⁹. These approaches directly model the physical processes governing ET using satellite-based observational constraints. Therefore, the products included in OpenET approximate total observed ET, including both natural and human contributions to ET. OpenET data have been validated against a benchmark eddy flux evapotranspiration dataset which includes a number of sites in California's Central Valley⁹⁵. Validation of OpenET data shows good performance across vegetation types at a monthly resolution, particularly across cropland sites ($R^2=0.9$)⁵². In this study, we use the OpenET ensemble estimate which has better performance than any individual ET model alone⁵².”

For NLDAS we have revised lines 458-477:

“NLDAS uses gauge-based precipitation, downward shortwave and longwave radiation, and surface meteorology reanalysis (10-m windspeed, 2-m air temperature, 2-m specific humidity, surface pressure) to drive three land surface models to produce outputs of naturally occurring surface fluxes (including ET), soil moisture, snow cover, and streamflow⁵⁰. Three models within NLDAS (Noah⁹⁶, Mosaic⁹⁷, and VIC⁹⁸) simulate the surface energy balance using the meteorological forcing data. Importantly, NLDAS models the natural processes contributing to ET, which does not include changes in ET based on human influence (e.g., irrigation)²⁷. Unlike OpenET, NLDAS does not include a land surface temperature constraint which would help distinguish natural ET based on the surface energy balance from total ET (including processes not considered in the model)⁵⁰. Validation of NLDAS models has been performed using spatial and temporal averaging over AmeriFlux stations to overcome spatial-scale incompatibility issues^{51,53}. NLDAS demonstrates good performance representing

ET across vegetation types⁵¹, and the lack of explicit representation of irrigation has been identified as a key source of error in the models⁵⁰. After evaluating the ability of each of the individual NLDAS land surface models to capture ET in our research domain using OpenET point data over non-irrigated regions, we averaged the VIC⁹⁸ and Mosaic⁹⁷ models as the highest performing combination given Noah's low predictive power over California⁵¹."

2) No evidence confirms NLDAS represents purely natural ET in human-modified landscapes.

NLDAS does not simulate irrigation or irrigation demand. This has been demonstrated in multiple previous publications (see Xia et al., 2012, Xia et al., 2015, Pascolini-Campbell et al., 2021 all cited in this study). To more clearly explain this, we have modified the text in lines 466-473:

"Unlike OpenET, NLDAS does not include a land surface temperature constraint which would help distinguish natural ET based on the surface energy balance from total ET (including processes not considered in the model)⁵⁰. Validation of NLDAS models has been performed using spatial and temporal averaging over AmeriFlux stations to overcome spatial-scale incompatibility issues^{51,53}. NLDAS demonstrates good performance representing ET across vegetation types⁵¹, and the lack of explicit representation of irrigation has been identified as a key source of error in the models⁵⁰."

3) Performance of both ET products were unclear, particularly in irrigation hotspots such as the central valley.

Both NLDAS and OpenET products have been tested and validated across a broad range of vegetation and climate types, both including California's central valley.

For NLDAS, Xia et al., 2014 and Zhang et al., 2020 evaluate the performance across multiple sites across the United States. For these analyses, the spatial mis-match between site-level data and gridded NLDAS data makes a direct comparison difficult. These studies overcome this by using spatial and temporal averaging across sites and time. We have better explained this including relevant references in lines 469-473:

"Validation of NLDAS models has been performed using spatial and temporal averaging over AmeriFlux stations to overcome spatial-scale incompatibility issues^{51,53}. NLDAS demonstrates good performance representing ET across vegetation types⁵¹, and the lack of explicit representation of irrigation has been identified as a key source of error in the models⁵⁰."

For an additional quality control check, given that prior validation had fewer California sites, we evaluated NLDAS against OpenET data over a few point locations of non-irrigated regions and show good agreement between NLDAS and OpenET. We explain this in lines 473-477:

“After evaluating the ability of each of the individual NLDAS land surface models to capture ET in our research domain using OpenET point data over non-irrigated regions, we averaged the VIC⁹⁸ and Mosaic⁹⁷ models as the highest performing combination given Noah’s low predictive power over California⁵¹.”

For OpenET, Volk et al., 2023 (cited in this study) use the following validation sites:

“The annual crop sites in the OpenET flux dataset are predominantly irrigated, and are distributed across a range of climatic zones, with higher density in regions such as Mediterranean and semi-arid Central Valley, California, and humid continental regions in the High Plains and the Mississippi Alluvial Plain (Fig. 1).”

Volk et al., 2024 show that OpenET shows good performance, particularly in cropland sites. We have modified our methods to more clearly demonstrate this in lines 447-451:

“OpenET data have been validated against a benchmark eddy flux evapotranspiration dataset which includes a number of sites in California’s Central Valley⁹⁵. Validation of OpenET data shows good performance across vegetation types at a monthly resolution, particularly across cropland sites ($R^2=0.9$)⁵².”

There are some specific comments listing as follows:

The numbers in front of the comments indicate line number.

1. L60. Incorrect citation “P W Liu” and similar ones across the manuscript.

This citation is to distinguish this publication from the other publications by J. Liu and X. Liu. The author of this paper is Pang-Wei Liu and the formatting occurs automatically using the Zotero citation manager in Word. We have changed citation formatting to be Nature standard citation format.

2. L69. ET is important. How about the basic water components in California. Such as annual rainfall, irrigated area, and how about agriculture water consumption. Is any study showing the relationship between ET and groundwater extraction?

We agree that annual rainfall, irrigated area, and agricultural water consumption are all important components of the water budget. We have defined the key components of the water budget in lines 60-64:

“To predict future water resources and water security, we need to understand how California’s changing climate system will impact different components of the water budget, including precipitation, water flow into and out of watersheds, water storage, and evapotranspiration (the sum of evaporation from soils and surface-water bodies and plant transpiration)¹¹.”

To our knowledge, no studies have directly linked ET to groundwater extraction in California. Liu et al., 2022 (cited in this study) show that groundwater depletion accelerates during drought in California's Central Valley, although this work does not provide an analysis of where the water goes once extracted.

3. L148. Inconsistency between NLDAS and NDLAS.

We have resolved this and checked for consistency throughout the manuscript.

4. L217. What does FHI represent? Please consider defining the abbreviation at the first time.

FHI is defined as "fractional human impact" the first time it appears in line 142.

5. L348. Is any text showing the change of crops?

These lines in the discussion are referencing previous work. We have modified this text for clarity.

Response to reviewers for “Human contributions to evapotranspiration mitigate swings in dry to wet year transitions”

We thank the editor and reviewers for their careful consideration of our manuscript. We have reviewed the feedback and offer a revised version.

We believe these changes have improved the manuscript and addressed concerns brought forth by reviewer 3. More details on these changes, including specific responses to reviewer concerns are found below italicized. We look forward to your re-consideration of our manuscript.

REVIEWER COMMENTS:

Editor:

Your revised manuscript titled "Human contributions to evapotranspiration mitigate swings in dry to wet year transitions" has now been seen by our reviewers, whose comments appear below. In light of their advice, we are delighted to say that we are happy, in principle, to publish a suitably revised version in Communications Sustainability,

We thank the editorial board for their enthusiasm for publishing a suitably revised version of this manuscript!

provided you acknowledge the limitation of NLDAS in explicit representation of irrigation and the uncertainty this introduces in evapotranspiration estimates, and tone down conclusions accordingly, along the line recommended by our reviewer 3.

We have added additional comments on the limitations of NLDAS for not explicitly representing irrigation in lines 488-491:

“NLDAS demonstrates good performance representing ET across vegetation types⁵¹, particularly over natural ecosystems with no known irrigation. Therefore, NLDAS does a good job representing naturally occurring ET fluxes. Prior validation work has shown that the lack of explicit representation of irrigation is a key source of error in the NLDAS models⁵⁰, making it unsuitable for representing total ET over irrigated areas.”

And revised our conclusions to reflect this in lines 436-437:

“These results show statistical significance notwithstanding uncertainties in OpenET and NLDAS datasets.”

Additional modifications and clarifications as requested by the reviewer are outlined in the response to Reviewer 3 below.

If you can address this request, we therefore invite you to revise your paper to comply with our format requirements and to maximise the accessibility and therefore the impact of your work.

Please see the attached response to the editorial requests.

Reviewer #3 (Remarks to the Author):

The authors have revised the manuscript. However, I remain concerned about the methodology for partitioning ET, specifically the derivation of "human ET" as the arithmetic difference between OpenET and NLDAS ET products.

The authors stated that "NLDAS does not simulate irrigation" and that this "lack of explicit irrigation representation is a key source of model error". This only confirms NLDAS's unsuitability for irrigated lands; it does not imply that ET over irrigated land represents solely natural water loss.

The reviewer is correct that NLDAS is not suitable for representing total ET over irrigated lands, as it does not consider the additional water input as a result of irrigation. NLDAS models the outputs of naturally occurring surface fluxes including ET as explained in lines 475-481:

"NLDAS uses gauge-based precipitation, downward shortwave and longwave radiation, and surface meteorology reanalysis (10-m windspeed, 2-m air temperature, 2-m specific humidity, surface pressure) to drive three land surface models to produce outputs of naturally occurring surface fluxes (including ET), soil moisture, snow cover, and streamflow⁵⁰. Three models within NLDAS (Noah⁹⁷, Mosaic⁹⁸, and VIC⁹⁹) simulate the surface energy balance using the meteorological forcing data. Importantly, NLDAS models the natural processes contributing to ET, which does not include changes in ET based on human influence (e.g., irrigation)²⁷."

To more clearly explain the suitability of NLDAS for capturing naturally occurring ET fluxes we have revised lines 487-493:

"NLDAS demonstrates good performance representing ET across vegetation types⁵¹, particularly over natural ecosystems with no known irrigation. Therefore, NLDAS does a good job representing naturally occurring ET fluxes. Prior validation work has shown that the lack of explicit representation of irrigation is a key source of error in the NLDAS models⁵⁰, making it unsuitable for representing total ET over irrigated areas."

Moreover, the algorithms adopted for estimating OpenET also do not explicitly simulate irrigation.

The reviewer is again correct that the algorithms for OpenET do not explicitly simulate irrigation. However, unlike NLDAS, they are constrained by satellite-based land surface temperature data which detects an actual evaporative response, which is then used to translate into an ET estimate. Therefore, OpenET models are agnostic to the input water source, as they are indirectly observing the ET response. We have modified the text to more accurately describe this in lines 462-466:

“These approaches directly model the physical processes governing ET using satellite-based observations of the evaporative response. Therefore, the products included in OpenET approximate total observed ET, including both natural and human contributions to ET, and are independent of blue or green water sources.”

While remote sensing data (used in OpenET) captured at an instant time might reflect irrigation effects, upscaling instantaneous ET to daily/monthly scales may not fully account for irrigation occurring later in the period, introducing uncertainty into monthly ET estimates. Although OpenET validation shows an R^2 of 0.9, the average normalized MAE (RMSE) was 17% (22%). Validation at orchard sites in the Central Valley, California, indicates similar uncertainty levels.

Upscaling instantaneous ET to daily/monthly scales may not fully account for irrigation that occurs at various times throughout the growing season. However, monthly ET tends to be more accurate than instantaneous/daily ET due to the cancellation of errors and reduced uncertainty in energy balance closure of validation data (Pierrat et al., 2025; Volk et al., 2023; Allen et al., 2007).

Furthermore, the RMSE of 17% is considered in Section 4.2 where we propagate the RMSE from OpenET and NLDAS through our estimates of human-ET. The resultant RMSE is 26 mm/month in cultivated crops. Our results show that despite this inherent error in the data sources used, there is still a statistically significant difference between human ET between 2022 and 2023 ($p < 0.05$ in a Mann-Whitney U test). We have modified the text to more appropriately explain this in lines 469-470:

“Validation of OpenET data shows good performance across vegetation types at a monthly resolution, particularly across cropland sites ($R^2=0.9$, Table 1 for RMSE) ⁵².”

And lines 566-572:

“For all our analysis we used the growing season average (May-September) to capture the peak of evapotranspiration and vegetation productivity in the years of interest. Furthermore, monthly data tends to be more accurate than daily or

instantaneous ET due to the cancellation of errors and reduced uncertainty in energy balance closure of validation data ^{52,103,104}. As a point of reference, we established a growing season average historical baseline by averaging the May-September monthly data from 2016-2021 for all datasets of interest.”

And lines 402-407:

“These known limitations do not undermine our main finding that dry-wet climatic swings do not significantly change total ET in California due to the strong human signature on ET, particularly in managed lands. Our results show significance despite known uncertainties in data sources, as the changes in human ET and FHI in managed lands are statistically significant and notably larger than the spread in the data (Table 1, Figure 4).”

The final decision remains with the editor. Nonetheless, the authors should explicitly acknowledge the limitations outlined above.

We have explicitly acknowledged the limitations outlined above as in the quoted revised text. Additionally, we have dedicated a full paragraph in the discussion to the limitations in these datasets in lines 374-401:

“The approach used in this study to delineate total, natural, and human ET processes has the potential to be used in other regions and across years to understand the water balance and future water resources. Despite this potential, there are several limitations to this work leading to potential errors in quantifying the relative contribution of human ET. These errors appear most notably as reported negative human ET and FHI in barren land, shrub/scrub, and grasslands as well as a non-zero FHI for evergreen needleleaf forests (Figure 4). We attribute these issues to two main limitations. First, known biases in the models themselves as a consequence of model forcing data errors, model structure deficiencies, model calibration errors ^{51,52}. OpenET has higher error in shrublands and grasslands compared with croplands ⁵². Despite this variability, OpenET does not show a bias in shrub and grassland ecosystems ⁵², giving us confidence that despite the large scatter in ET values, the statewide average will converge on the true value. This is consistent with our results which show an average of 0 mm month⁻¹ human ET and 0% FHI in barren land, shrub/scrub, and grasslands, despite notable spread in the data. In evergreen needleleaf forests, OpenET has a known high bias ⁵², and NLDAS has a low bias ⁵¹. Taken together, this leads to an over-inflation of human ET in evergreen needleleaf systems, consistent with our results. Second, the coarse resolution of NLDAS requires spatial aggregation of multiple land cover types and may not adequately capture fine scale variability in natural and human ET impacts present in the OpenET data. This spatial aggregation is particularly relevant when considering fine-scale changes in agricultural regions such as crop rotation and fallowing. Future work should consider potential downscaling approaches for understanding more water efficient irrigation practices ²⁶. The aggregation of multiple land cover types in a

single pixel may also contribute to the non-zero FHI in areas which would not be expected to have an irrigation/human influence (e.g., evergreen forests) if irrigated regions are within the pixel. However, the spread in the data was less than the expected RMSE based on previous work, suggesting that the spatial aggregation is less important than improving the data products themselves.”

And detail previously reported uncertainties in the datasets in Section 4.

**Irrigation contributions to evapotranspiration mitigate swings in dry to wet year**
**transitions**

Zoe Amie Pierrat¹, Rebecca N. Gustine^{1,3}, Anna Boser², Sophie Ruehr⁴, Christine M.
Lee¹, J. T. Reager¹, Kerry Cawse-Nicholson¹

¹ NASA Jet Propulsion Laboratory, California Institute of Technology, Pasadena, CA,
91011

² Bren School of Environmental Science & Management, University of California, Santa
Barbara, Santa Barbara, CA, 93106

³ Lamont-Doherty Earth Observatory of Columbia University, Palisades, NY, 10964

⁴ Department of Environmental Science Policy and Management, University of
California Berkeley, Berkeley, CA, 94720

Corresponding authors: Zoe Amie Pierrat (zoe.a.pierrat@nasa.jpl.gov) and Christine M.
Lee (christine.m.lee@nasa.jpl.gov)

**Key Points:**

- - Comparing observed ET with modeled ET reveals how dry-wet year transitions
like those exhibited between 2022 to 2023 in California, impact total ET, natural
ET, and the human impact on ET.
- Total ET changes <10% between dry-wet year transitions indicate that human
activity increases ET demand during dry years.
- Irrigation contributions to ET in managed lands range from ~80% in dry years to
~50% in wet years, underscoring the need for efficient irrigation practices for
water resources sustainability.

**Abstract**

California's food and economic security depends on water availability and
regional water fluxes, particularly under extreme climate scenarios. Evapotranspiration
(ET) is a function of natural (e.g., climate, vegetation cover) and human controls (e.g.,

irrigation), both of which impact water availability. In this study, we analyze the transition
from one of California's driest years (2022) to an exceptionally wet year (2023) to
determine how natural and human-driven ET each respond to these natural climate
fluctuations. Between the two years, total ET remained constant (<10% change) despite
the large change in precipitation. In the dry year, evapotranspiration contributed by
irrigation represented 30% of the statewide ET signal and 80% in managed lands. In the
wet year, natural ET increased, resulting in a -30% reduction in irrigation-related ET and
minimal change in total ET. Nevertheless, in 2023, the fractional irrigation ET remained
nearly 50% in managed lands. Our findings underscore a persistent reliance on
irrigation in the Central Valley, even during wet years. As climate change intensifies
hydroclimatic variability, applying our framework for understanding the interplay
between natural and human contributions to ET will aid in water resource planning in
California and other water-limited regions.

**Plain Language Summary**

**1. Introduction**

California generates more than \$50 billion in annual revenue from the agricultural
sector, predominantly from irrigated crops (Peterson et al., 2022). Over a third of the
vegetable and over three-quarters of the fruits and nut production for the United States
comes from California (California Agricultural Exports, 2022). Despite more efficient
irrigation systems in agricultural land, consumptive water use by humans has remained
stable (Peterson et al., 2023). The stability in consumptive water use can be explained
by California's 10% increase in agricultural output since 2013 (California Agricultural
Exports, 2022), California's changing environmental conditions including increased
frequency and severity of drought (Liu et al., 2022; Williams et al., 2020), and more
extreme swings from dry to wet conditions (Swain et al., 2016, 2025). California's
increased agricultural output and changing environmental conditions impact rates of
evapotranspiration (ET) across the state, which can increase irrigation demand, deplete
groundwater storage and ultimately impact future water security. Therefore,

understanding changes to ET across California is critical for protecting the economic
and agricultural prosperity of the state.

ET is the combination of water *evaporation* through soil and surface water bodies
and *transpiration* through plant stomata. ET is an important part of the hydrologic cycle
and a critical component of sustainable water resources decision making, particularly in
agriculture (Ghiat et al., 2021; Miralles et al., 2020; Novák, 2011; Wanniarachchi &
Sarukkalige, 2022; Ward, 1971). The rate of ET is controlled by environmental factors
including temperature, soil moisture, solar radiation, wind, and atmospheric vapor
pressure (Bento et al., 2018; Fisher et al., 2008; Joiner et al., 2018); biotic factors like
ecosystem type, vegetation health, and plant functional traits (Bhattarai & Wagle, 2021;
Brown et al., 2010; Detto et al., 2006); and water applied via irrigation (Boser et al.,
2024; Pascolini-Campbell et al., 2021). Accordingly, ET is a combination of both natural
(environmental controls and vegetation health) and human (management and irrigation)
contributions. Understanding how both natural and human contributions to the ET signal
change in California because of changing environmental conditions will help us
anticipate and manage future changes to the water budget.

Climatically, seasonal and interannual swings from dry to wet conditions have
amplified in the western US (Allan, 2023; Rodell & Li, 2023). However, the subsequent
impacts of these types of swings from extreme dry to extreme wet conditions on the ET
response and subsequent water budget in California is unknown. Under drought
conditions, natural and human contributions to ET frequently increase due to higher
evaporative demand (Zhao et al., 2022). This higher evaporative demand then
increases irrigation needs and can lead to an accelerated depletion of groundwater
storage (Liu et al., 2022). On the other hand, ET can also be reduced under extreme
drought, driven by a decline in available water sources for soil evaporation and plant
transpiration, as well as vegetation mortality (Seneviratne et al., 2010; He et al., 2022).
Changes to ET under drought conditions thus depend on the duration, severity, and
intensity of the drought/dry period, and human influence. Enhanced precipitation and
total water storage anomalies following drought can improve vegetation recovery and
rebalancing of the ET signal (Au et al., 2023; Zhao et al., 2022). A vegetation boom
following high precipitation may also contribute to an enhanced risk of wildfire, further

altering the water budget (Ma et al., 2020; Swain, 2021). Therefore, understanding the
ET response to climatic swings in California, which are expected to become more
extreme and more frequent, will improve our understanding of the regional water
balance, including groundwater recharge and future water resource availability
(DeFlorio et al., 2024).

Here, we compare how vegetation and human processes in an exceptionally wet
95 year (2023) and an exceptionally dry year (2022) contributed to changes in
evapotranspiration and subsequent water demand in California. Specifically, we ask, 1)
how did evapotranspiration change in California from an extreme dry year (2022) to a
record-breaking water year (2023)? and 2) how did the relative contributions of natural
and human influences on the ET signal contribute to or mitigate those changes? To
answer these questions, we use observationally constrained estimates of ET from
OpenET to represent total ET, modeled ET from the National Land Data Assimilation
System (NLDAS) to represent natural contributions to ET, and the difference between
them to represent human contributions to ET. Using these cutting-edge data and
modeling approaches, we shed light on how human consumptive water use and overall
water resources may shift under a changing climate. This study has implications for
sustainable water management in California, the United States' largest agricultural
producer, and our methodological approach shows potential for understanding future
changes to ET around the world.

**2. Results**

California is dominated by shrub/scrub (42.7% of regrided pixels), evergreen
forest (17.6%), grassland/herbaceous (14.9%), cultivated crops (9.9%), and developed,
medium intensity land (2.3%) (Figure 1a). Growing season baseline ET data from
OpenET (Total ET), NLDAS (Natural ET), and the difference between them (Human
ET), reveal the unique spatial patterns and variable contributions to ET across California
(Figure 1 b-d). Growing season baseline (2016-2021) *total* ET is an average of 45 ± 29
116 mm/month across the state. The majority of this comes from northern and coastal
California (where evergreen forests dominate) and the Central Valley (which is
cultivated crops). The highest average growing season baseline (2016-2021) total ET

rates (Figure 1b) are found in evergreen forests (79 ± 18 mm/month), followed by
cultivated crops (69 ± 18 mm/month), developed, medium intensity land (60 ± 11
121 mm/month), grassland/herbaceous (34 ± 19 mm/month), shrub/scrub (33 ± 22
122 mm/month), and barren land (14 ± 11 mm/month). Natural ET is the largest contributor
to the total ET signal with an average growing season baseline of 32 ± 21 mm/month
(71% of total) across the state (Figure 1c). The highest ET rates come from evergreen
forests (60 ± 11 mm/month), followed by grassland/herbaceous (36 ± 18 mm/month),
shrub/scrub (25 ± 17 mm/month), cultivated crops (19 ± 11 mm/month), developed,
medium intensity (16 ± 5 mm/month), and barren land (13 ± 15 mm/month).

Human ET contributes an average of 13 ± 23 mm/month (29% of total) to the
total ET signal (Figure 1d). The highest average growing season baseline human ET
rates come from cultivated crops (49 ± 20 mm/month), followed by developed, medium
intensity (43 ± 15 mm/month), evergreen forests (19 ± 18 mm/month), shrub/scrub ($7 \pm$
18 mm/month), barren land (1 ± 11 mm/month), and grassland/herbaceous (-2 ± 16
133 mm/month). Baseline FHI is highest in the Central Valley and along southern coastal
California, where cultivated crops and developed land dominate and humans have a
strong influence on ET rates (Figure 1e). In these managed areas, developed land has
the highest FHI at $71 \pm 16\%$, closely followed by cultivated crops with a FHI of $70 \pm$
19% . Baseline NDVI reflects the state of vegetation cover across California (Figure 1f).
NDVI is highest in evergreen forests (0.67 ± 0.12), followed by cultivated crops ($0.46 \pm$
0.09), grassland/herbaceous (0.42 ± 0.15), developed, medium intensity (0.31 ± 0.06),
shrub/scrub (0.30 ± 0.18), and barren land (0.12 ± 0.06). Taken together, these results
highlight the strong impact of human processes on total ET across California, with
human contributions to the ET signal being comparable or greater than natural ET
across much of the state.

*Figure 1: a) Landcover classifications across California for 2023. The baseline (2016-*
 *2021) summer (May-September) ET rates for b) total ET from OpenET, c) natural ET*
 *from NLDAS, d) human ET as the difference between OpenET and NLDAS ET, as well*
 *as e) FHI, and f) NDVI.*

Comparing total, natural, and human average ET rates in 2022 and 2023 clarifies
 the impact of climatic swings on ET across California. Despite being an exceptionally
 dry year in 2022, California experienced an average total ET rate of 49 ± 28 mm/month

(Figure 2a). This represented an average increase in ET of 16% relative to the 2016-
2021 baseline. In 2023, the exceptionally wet year, California experienced an average
total ET rate of 52 ± 26 mm/month, a 35% increase relative to the 2016-2021 baseline
(Figure 2b). From 2022 to 2023, California experienced an average increase in ET rates
by 3 ± 9 mm/month (6% increase from 2022) (Figure 2c). Despite the relatively small
average increase in ET across the state, changes in ET rates were highly non-uniform,
and changes in ET rates were mediated by changes to natural vs. human contributions
to the ET signal. Natural ET contributed an average of 29 ± 1 mm/month (59% of total
ET) in 2022 (Figure 2d), while in 2023, natural ET contributed an average of 47 ± 23
162 mm/month (90% of total ET) (Figure 2e). Between 2022 and 2023, natural ET increased
by an average of 17 ± 13 mm/month (Figure 2f). Human ET contributed an average of
20 ± 24 mm/month (41% of total ET) in 2022 (Figure 2g) and 5 ± 21 mm/month (10% of
total ET) in 2023 (Figure 2h). Between 2022 and 2023, human ET decreased by an
average of -14 ± 14 mm/month (Figure 2i). FHI in 2022 (Figure 2j) and 2023 (Figure 2k)
was highest in the Central Valley and southern California and exhibited an average
decrease across the state of $-31 \pm 39\%$ from 2022 to 2023 (Figure 2l). Despite this
overall decrease, a high FHI impact remained in the Central Valley and Los Angeles in
2022. Average NDVI was 0.38 ± 0.20 in 2022 (Figure 2m) and 0.41 ± 0.20 in 2023
(Figure 2n), representing an average increase of 0.03 (8%) (Figure 2o).

*Figure 2: Growing season (May-September) evapotranspiration in the dry year (2022,*

*left column), the wet year (2023, middle column), and the difference between them*

*(right column) for a-c) total ET from OpenET, d-f) natural contributions to ET from*

*NLDAS, g-i) human contributions to ET as the difference between OpenET and NLDAS*

*ET, j-l) the fractional human impact (FHI) on ET, and m-o) NDVI.*

The changes to ET rates and ultimate impacts to the water balance become
clearer when breaking up changes in ET, FHI, and NDVI by landcover classification
(Figure 3). Between 2022 and 2023, natural land cover classifications (shrub/scrub,
grassland/herbaceous, evergreen forests) showed small but statistically significant
changes to ET, while managed lands (developed land, cultivated crops) showed no
significant change in ET (Figure 3a). Natural ET, which shows a clear increase in ET
rates in 2023 across the board: shrub/scrub (23 ± 16 mm/month in 2022 to 39 ± 21
186 mm/month in 2023), grasslands (29 ± 22 mm/month in 2022 to 54 ± 17 mm/month in
2023), developed land (12 ± 6 mm/month in 2022 to 30 ± 9 mm/month in 2023), and
crops (12 ± 9 mm/month in 2022 to 37 ± 15 mm/month in 2023) (Figure 3b). Human ET
shows a smaller change between years for land cover types which are not expected to
have a human impact (barren land, shrub/scrub, grasslands, evergreen forests) while
developed land and cultivated crops show a clear decrease in human ET in 2023 ($44 \pm$
12 mm/month in 2022 to 25 ± 16 mm/month in 2023 for developed land; and 58 ± 19
193 mm/month in 2022 to 36 ± 21 mm/month in 2023 for cultivated crops) (Figure 3c).
Consequently, the fractional human impact on ET changed significantly for developed
land ($78 \pm 12\%$ in 2022 to $41 \pm 25\%$ in 2023) and cultivated crops ($80 \pm 15\%$ in 2022 to
$47 \pm 27\%$ in 2023). NDVI shows small changes between the two years, with the largest
increases in NDVI occurring in grassland/herbaceous (0.37 ± 0.09 in 2022 to $0.42 \pm$
0.09 in 2023) followed by cultivated crops (0.45 ± 0.09 in 2022 to 0.48 ± 0.09 in 2023).

*Figure 3: The distribution of evapotranspiration broken up by land cover and separated*
 *by year (dry 2022 vs. wet 2023) for a) total ET, b) natural ET, c) human ET, d) FHI, e)*
 *NDVI. The red stars (*) indicate groupings where there was a statistically significant*
 *difference ($p < 0.05$ in a Mann-Whitney U test).*

**3. Discussion**

With this work, we clarify how evapotranspiration changed in California during a
“record breaking water year” (2023) following a record dry year (2022) and how the
relative contributions of natural and human influences on the ET signal mediated those
changes. Overall, ET rates show only a small increase in the high water year (average
<10% change), despite the large increase in precipitation (141% of statewide average,
(California Department of Water Resources et al., 2023)). The stable ET signal can be
explained by the changing contributions of natural and human processes, and
demonstrates the environmental (temperature, humidity, natural disturbances such as
fires or pests) and human controls (irrigation, crop type) on ET (Brown et al., 2010;
Detto et al., 2006; Ma et al., 2020; McDonald & Girvetz, 2013; Rajagopalan et al.,
2018).

High temperatures and extended droughts increase atmospheric evaporative
demand and plant water stress (Zhang et al., 2015; Zhao et al., 2022) resulting in an
increased need for irrigation (Cook et al., 2020; Döll & Siebert, 2002) and a resulting
depletion of groundwater storage (Liu et al., 2022) due to groundwater-driven irrigation.
These conditions were present across California from 2020-2022 (Janes, 2024), which
led to the high FHI (80%) in the dry year (2022) and the prevention of changes to total
ET in the wet year (2023). Despite a reduction in human ET in 2023, the FHI on
managed lands (developed and cultivated crops) remained high (nearly 50%),
suggesting that future “boom water years” will still be insufficient for the water demands
of California’s agricultural sector, putting the future of California water and economic
security at risk. Drought conditions are expected to increase through the 21st century,
which further underscores the need for water-smart agricultural practices (Boser et al.,
2024; Fader et al., 2016).

There are several approaches for reducing the human-driven contribution to total
ET. Recent work by Boser et al. (2024) has demonstrated a potential 10% reduction in
water consumption because of 1) crop selection (from high-ET crops to lower ET-
crops), 2) farming practices (e.g., reduced irrigation to the same crops), or c) following

5% of land. This work also noted a maximum 94% reduction in agricultural ET as a
result of switching to lowest water use crops. Water rights and water outlooks (United
States Department of Agriculture Natural Resources Conservation Service, 2023) affect
which annual crops get planted and subsequently, affect yearly ET values in managed
lands (Boser et al., 2024; Gebremichael et al., 2021; Nelson & Burchfield, 2017). Our
results show an increase in NDVI in cultivated crops in 2023, which may reflect a
change in planted crops to more water-demanding crops following a high water outlook
for 2023 (Agrawal et al., 2021; Escriva-Bou et al., 2022; Guido et al., 2020; Peterson et
al., 2022), or a decrease in the amount of fallowed land (Boser et al., 2024; Peterson et
al., 2023). Our results underscore the need for water-smart switches, even in years with
high water outlooks.

In natural areas increased water availability, like that observed in 2023, can lead
to increased ET rates due to increased vegetation health and abundance. Grasslands,
barren land, and shrub/scrubland all showed subtle increases in NDVI due to higher
water availability in 2023. This led to strong increases in ET rates in these areas as
transpiration increased (Bento et al., 2018). In contrast, evergreen forests showed
smaller changes in NDVI and ET between 2022 and 2023. NDVI is a poor predictor of
productivity in evergreen forests (Pierrat et al., 2024); therefore, NDVI may not
accurately capture interannual variability in these ecosystems. The fact that ET also
remained relatively consistent in evergreen forests between years suggests a higher
resilience to extreme swings in precipitation from year to year. This is likely because
evergreen forests are able to access deeper groundwater stores, making groundwater
recharge in these areas following extreme precipitation critical for the sustained
resilience of these ecosystems (Au et al., 2023). Multi-year droughts can deplete deep
groundwater reserves and lead to forest die-off (Goulden & Bales, 2019), emphasizing
the need for groundwater recharge during high water years.

The effect of the hydroclimate volatility on vegetation and ET in California as
observed in 2022 and 2023 will have cascading consequences for the natural
ecohydrology of California, namely wildfire regimes (Swain, 2021; Swain et al., 2025).
When a wet year follows a dry year, the increased water and nutrient availability can

lead to a boom in vegetation growth, particularly in mediterranean climates and semi-
arid shrubland ecosystems (Hernández Ayala et al., 2021), consistent with what was
observed in this study. This boom in vegetation growth increases fuel loading and
subsequently, increased wildfire severity (Farahmand, et al., 2020; Farahmand, et al.,
2020; Jensen et al., 2018). Climate change is amplifying this wet to dry trend by
intensifying both jet streams and Santa Ana winds (Guirguis et al., 2023). This pattern
was observed in the severe California wildfire season in 2020, following 2019 being the
first non-drought year in California since 2011 (Cal Fire and Watch Duty). The 2024 fire
season initially followed this trend, burning more acreage than the following 5 year-
average in the early summer months (Toohey, 2024), and may have contributed to the
extreme fire events in Los Angeles in early 2025 (Cal Fire and Watch Duty). Future
research should investigate the connections to the 2024-2025 fire season.

The approach used in this study to separate out total, natural, and human ET
processes has the potential to be used in other regions and across years to understand
the water balance and future water resources. Despite this, there are several limitations
to this work leading to potential errors in quantifying the relative contribution of human
ET. These errors show up most notably as reported negative human ET rates and FHI
in barren land, shrub/scrub, and grasslands as well as a non-zero FHI for evergreen
needleleaf forests (Figure 3). We attribute these issues to two main limitations. First,
known biases in the models themselves as a consequence of model forcing data errors,
model structure deficiencies, model calibration errors (Volk et al., 2024; Xia et al.,
2015). OpenET has higher error in shrublands and grasslands compared with croplands
(Volk et al., 2024). Despite this variability, OpenET does not show a bias in shrub and
grassland ecosystems (Volk et al., 2024), giving us confidence that despite the large
scatter in ET values, the statewide average will converge on the true value. This is
consistent with our results which show an average of 0 mm/month human ET and 0%
FHI in barren land, shrub/scrub, and grasslands, despite notable spread in the data. In
evergreen needleleaf forests, OpenET has a known high bias (Volk et al., 2024), and
NLDAS has a low bias (Xia et al., 2015). Taken together, this leads to an over-inflation
of the human ET signal in evergreen needleleaf systems, consistent with our results.
Second, the coarse resolution of NLDAS requires spatial aggregation of multiple land

cover types and may not adequately capture fine scale variability in natural and human
ET impacts present in the OpenET data. This is particularly relevant when considering
fine-scale changes in agricultural regions such as crop rotation and fallowing and future
work should consider potential downscaling approaches for understanding more water
efficient irrigation practices (Boser et al., 2024). The aggregation of multiple land cover
types in a single pixel may also contribute to the non-zero FHI in areas which would not
be expected to have an irrigation/human influence (e.g., evergreen forests) if irrigated
regions are within the pixel. These known limitations do not undermine our main finding
that climatic swings from dry to wet do not significantly change the total ET signal in
California due to the strong human signature on ET, particularly in managed lands.
Ultimately, the changes in human ET and FHI in managed lands in this study are
statistically significant and notably larger than the spread in the data, giving us
confidence that despite potential uncertainties, our approach demonstrates the ability to
capture changing contributions to ET from human and natural processes as a
consequence of extreme climatic swings (Xia et al., 2014).

**4. Methods**

To quantify the differences between ET response to climatic swings from
exceptionally dry to exceptionally wet, we used a combination of different datasets on
land cover, vegetation health, and evapotranspiration. The years 2022-2023 in
California provide an ideal natural case study. In 2023, California was hit by repeated
Pacific jet streams bringing extreme precipitation, snowfall, and increased snowpack
levels compared with the historical baseline, creating a “record breaking water year”
(US National Park Service, 2023). This record water year was preceded in 2022 by
some of the most extreme drought conditions in California history. These two years
allow us to test the response of vegetation to extreme interannual swings and minimize
the impact of land cover change between the two years.

**4.1. Datasets**

To evaluate how changes in precipitation and available water impact different
land cover types, we used land cover classifications from the National Land Cover

Database (NLCD) (United States Geological Survey, 2023). NLCD data are available
over the conterminous United States from 1985-2023 at a 30-meter resolution. Open
water classes were excluded from this analysis but are still included in maps for
visualization.

To quantify the rates of *total* observed ET across the state of California, we used
monthly ET data from OpenET (Melton et al., 2022). OpenET data are available at a
30m x 30m resolution 2016-present. OpenET data are produced using an ensemble
estimate of 6 different ET models driven by satellite-based land surface temperature
and meteorological data from Landsat, Terra/Aqua MODIS, Suomi NPP VIIRS, GOES,
and Sentinel-2 (Melton et al., 2022; Volk et al., 2024a). Because OpenET is
observationally constrained with land surface temperature and meteorological data, it
represents *total* rates of ET including both natural and human contributions to ET.

To assess the *natural* contributions to ET (not including contributions due to
irrigation), we used monthly modeled ET data from the National Land Data Assimilation
System (NLDAS) which are provided at a 0.125 x 0.125-degree resolution from 1979-
present (Xia et al., 2012a; 2012b). NLDAS uses gauge-based precipitation, bias-
corrected shortwave radiation, and surface meteorology reanalysis to drive three land
surface models to produce outputs of naturally occurring surface fluxes (including ET),
soil moisture, snow cover, and streamflow. Importantly, NLDAS models the *natural*
processes contributing to the ET signal, which does not include changes in ET based on
human influence (e.g., irrigation) (Pascolini-Campbell et al., 2021). After evaluating the
ability of each of the individual NLDAS land surface models (i.e., Noah, Mosaic, and
VIC) to capture ET in our research domain using OpenET point data, we averaged the
VIC and Mosaic models as the highest performing combination given Noah's low
predictive power over California. We do not include the evaluation here but see Xia et
al. (2015) for a more comprehensive NLDAS ET evaluation.

We inferred *human* impact on evapotranspiration by taking the difference
between OpenET (i.e., total ET) and NLDAS ET (i.e., natural ET) (Human ET =
OpenET-NLDAS ET) (Pascolini-Campbell et al., 2021). We define fractional human
impact (FHI) as:

$$\text{FHI} = \frac{\text{Human ET}}{\text{Total ET}} \times 100\%$$

To assess overall changes in vegetation coverage as a key driver of the ET
signal, we used normalized difference vegetation index (NDVI) data from the NASA
Visible Infrared Imaging Radiometer Suite (VIIRS) Land Program (Obata et al., 2013;
Vermote, 2023). Data are provided monthly on a 1 km x 1 km grid and extend back to
January 2012.

**4.2. Data Processing**

[revised manuscript text omitted]

India. *Current Research in Environmental Sustainability*, 3, 100068.
<https://doi.org/10.1016/j.crsust.2021.100068>

Au, J., Bloom, A. A., Parazoo, N. C., Deans, R. M., Wong, C. Y. S., Houlton, B. Z., &
Magney, T. S. (2023). Forest productivity recovery or collapse? Model-data
integration insights on drought-induced tipping points. *Global Change Biology*,
29(19), 5652–5665. <https://doi.org/10.1111/gcb.16867>

Bento, V. A., Gouveia, C. M., DaCamara, C. C., & Trigo, I. F. (2018). A climatological
assessment of drought impact on vegetation health index. *Agricultural and Forest
Meteorology*, 259, 286–295. <https://doi.org/10.1016/j.agrformet.2018.05.014>

Bhattarai, N., & Wagle, P. (2021). Recent Advances in Remote Sensing of
Evapotranspiration. *Remote Sensing*, 13(21), Article 21.
<https://doi.org/10.3390/rs13214260>

Boser, A., Caylor, K., Larsen, A., Pascolini-Campbell, M., Reager, J. T., & Carleton, T.
(2024). Field-scale crop water consumption estimates reveal potential water
savings in California agriculture. *Nature Communications*, 15(1), 2366.
<https://doi.org/10.1038/s41467-024-46031-2>

Brown, S. M., Petrone, R. M., Mendoza, C., & Devito, K. J. (2010). Surface vegetation
controls on evapotranspiration from a sub-humid Western Boreal Plain wetland.
*Hydrological Processes*, 24(8), Article 8. <https://doi.org/10.1002/hyp.7569>

California Agricultural Exports. (2022). *California Agricultural Exports 2022-2023*.
[https://www.cdfa.ca.gov/Statistics/PDFs/2022-
2023_california_agricultural_exports.pdf](https://www.cdfa.ca.gov/Statistics/PDFs/2022-2023_california_agricultural_exports.pdf)

California Department of Water Resources, California Natural Resources Agency, &

State of California. (2023). *Water Year 2023: Weather Whiplash, From Drought*
*to Deluge*. [https://water.ca.gov/-/media/DWR-Website/Web-Pages/Water-](https://water.ca.gov/-/media/DWR-Website/Web-Pages/Water-Basics/Drought/Files/Publications-And-Reports/Water-Year-2023-wrap-up-brochure_01.pdf)
[Basics/Drought/Files/Publications-And-Reports/Water-Year-2023-wrap-up-](https://water.ca.gov/-/media/DWR-Website/Web-Pages/Water-Basics/Drought/Files/Publications-And-Reports/Water-Year-2023-wrap-up-brochure_01.pdf)
[brochure_01.pdf](https://water.ca.gov/-/media/DWR-Website/Web-Pages/Water-Basics/Drought/Files/Publications-And-Reports/Water-Year-2023-wrap-up-brochure_01.pdf)

Cook, B. I., McDermid, S. S., Puma, M. J., Williams, A. P., Seager, R., Kelley, M.,
Nazarenko, L., & Aleinov, I. (2020). Divergent Regional Climate Consequences
of Maintaining Current Irrigation Rates in the 21st Century. *Journal of*
*Geophysical Research: Atmospheres*, 125(14), e2019JD031814.
<https://doi.org/10.1029/2019JD031814>

DeFlorio, M. J., Sengupta, A., Castellano, C. M., Wang, J., Zhang, Z., Gershunov, A.,
Guirguis, K., Niño, R. L., Clemesha, R. E. S., Pan, M., Xiao, M., Kawzenuk, B.,
Gibson, P. B., Scheftic, W., Broxton, P. D., Switanek, M. B., Yuan, J., Dettinger,
448 M. D., Hecht, C. W., ... Anderson, M. L. (2024). *From California's Extreme*
*Drought to Major Flooding: Evaluating and Synthesizing Experimental Seasonal*
*and Subseasonal Forecasts of Landfalling Atmospheric Rivers and Extreme*
*Precipitation during Winter 2022/23*. <https://doi.org/10.1175/BAMS-D-22-0208.1>

Detto, M., Montaldo, N., Albertson, J. D., Mancini, M., & Katul, G. (2006). Soil moisture
and vegetation controls on evapotranspiration in a heterogeneous Mediterranean
ecosystem on Sardinia, Italy. *Water Resources Research*, 42(8), Article 8.
<https://doi.org/10.1029/2005WR004693>

Döll, P., & Siebert, S. (2002). Global modeling of irrigation water requirements. *Water*
*Resources Research*, 38(4), Article 4. <https://doi.org/10.1029/2001WR000355>

Escriva-Bou, A., Medellín-Azuara, J., Hanak, E., Abatzoglou, J., & Viers, J. (2022).

*Drought and California's Agriculture.*

Fader, M., Shi, S., von Bloh, W., Bondeau, A., & Cramer, W. (2016). Mediterranean
irrigation under climate change: More efficient irrigation needed to compensate
for increases in irrigation water requirements. *Hydrology and Earth System*
*Sciences*, 20(2), Article 2. <https://doi.org/10.5194/hess-20-953-2016>

Farahmand, A., Stavros, E. N., Reager, J. T., & Behrangi, A. (2020). Introducing
Spatially Distributed Fire Danger from Earth Observations (FDEO) Using
Satellite-Based Data in the Contiguous United States. *Remote Sensing*, 12(8),
Article 8. <https://doi.org/10.3390/rs12081252>

Farahmand, A., Stavros, E. N., Reager, J. T., Behrangi, A., Randerson, J. T., & Quayle,
B. (2020). Satellite hydrology observations as operational indicators of forecasted
fire danger across the contiguous United States. *Natural Hazards and Earth*
*System Sciences*, 20(4), 1097–1106. [https://doi.org/10.5194/nhess-20-1097-](https://doi.org/10.5194/nhess-20-1097-2020)
2020

Fisher, R. A., Williams, M., de Lourdes Ruivo, M., de Costa, A. L., & Meir, P. (2008).
Evaluating climatic and soil water controls on evapotranspiration at two
Amazonian rainforest sites. *Agricultural and Forest Meteorology*, 148(6), Article
6. <https://doi.org/10.1016/j.agrformet.2007.12.001>

Gebremichael, M., Krishnamurthy, P. K., Ghebremichael, L. T., & Alam, S. (2021). What
Drives Crop Land Use Change during Multi-Year Droughts in California's Central
Valley? Prices or Concern for Water? *Remote Sensing*, 13(4), Article 4.
<https://doi.org/10.3390/rs13040650>

Goulden, M. L., & Bales, R. C. (2019). California forest die-off linked to multi-year deep

soil drying in 2012–2015 drought. *Nature Geoscience*, 12(8), 632–637.
<https://doi.org/10.1038/s41561-019-0388-5>

Guido, Z., Zimmer, A., Lopus, S., Hannah, C., Gower, D., Waldman, K., Krell, N.,
Sheffield, J., Caylor, K., & Evans, T. (2020). Farmer forecasts: Impacts of
seasonal rainfall expectations on agricultural decision-making in Sub-Saharan
Africa. *Climate Risk Management*, 30, 100247.
<https://doi.org/10.1016/j.crm.2020.100247>

Hernández Ayala, J. J., Mann, J., & Grosvenor, E. (2021). Antecedent Rainfall,
Excessive Vegetation Growth and Its Relation to Wildfire Burned Areas in
California. *Earth and Space Science*, 8(9), Article 9.
<https://doi.org/10.1029/2020EA001624>

Janes, M. (2024). *California's Groundwater Conditions: Semi-Annual Update May 2024*.

Jensen, D., Reager, J. T., Zajic, B., Rousseau, N., Rodell, M., & Hinkley, E. (2018). The
sensitivity of US wildfire occurrence to pre-season soil moisture conditions
across ecosystems. *Environmental Research Letters*, 13(1), 014021.
<https://doi.org/10.1088/1748-9326/aa9853>

Joiner, J., Yoshida, Y., Anderson, M., Holmes, T., Hain, C., Reichle, R., Koster, R.,
Middleton, E., & Zeng, F.-W. (2018). Global relationships among traditional
reflectance vegetation indices (NDVI and NDII), evapotranspiration (ET), and soil
moisture variability on weekly timescales. *Remote Sensing of Environment*, 219,
339–352. <https://doi.org/10.1016/j.rse.2018.10.020>

Liu, P.-W., Famiglietti, J. S., Purdy, A. J., Adams, K. H., McEvoy, A. L., Reager, J. T.,
Bindlish, R., Wiese, D. N., David, C. H., & Rodell, M. (2022). Groundwater

depletion in California's Central Valley accelerates during megadrought. *Nature*
*Communications*, 13(1), 7825. <https://doi.org/10.1038/s41467-022-35582-x>

507 Ma, Q., Bales, R. C., Rungee, J., Conklin, M. H., Collins, B. M., & Goulden, M. L.
(2020). Wildfire controls on evapotranspiration in California's Sierra Nevada.
*Journal of Hydrology*, 590, 125364. <https://doi.org/10.1016/j.jhydrol.2020.125364>

McDonald, R. I., & Girvetz, E. H. (2013). Two Challenges for U.S. Irrigation Due to
Climate Change: Increasing Irrigated Area in Wet States and Increasing Irrigation
Rates in Dry States. *PLOS ONE*, 8(6), Article 6.
<https://doi.org/10.1371/journal.pone.0065589>

Nelson, K. S., & Burchfield, E. K. (2017). Effects of the Structure of Water Rights on
Agricultural Production During Drought: A Spatiotemporal Analysis of California's
Central Valley. *Water Resources Research*, 53(10), 8293–8309.
<https://doi.org/10.1002/2017WR020666>

Pascolini-Campbell, M., Fisher, J. B., & Reager, J. T. (2021). GRACE-FO and
ECOSTRESS Synergies Constrain Fine-Scale Impacts on the Water Balance.
*Geophysical Research Letters*, 48(15), e2021GL093984.
<https://doi.org/10.1029/2021GL093984>

Peterson, C., Escriva-Bou, A., Medellín-Azuara, J., & Cole, S. (2023). *Water Use in*
*California's Agriculture*. [https://www.ppic.org/wp-content/uploads/water-use-in-](https://www.ppic.org/wp-content/uploads/water-use-in-californias-agriculture.pdf)
[californias-agriculture.pdf](https://www.ppic.org/wp-content/uploads/water-use-in-californias-agriculture.pdf)

Peterson, C., Pittelkow, C., & Lundy, M. (2022). *Exploring the Potential for Water-*
*Limited Agriculture in the San Joaquin Valley*.

Pierrat, Z. A., Magney, T. S., Cheng, R., Maguire, A. J., Wong, C. Y. S., Nehemy, M. F.,

Rao, M., Nelson, S. E., Williams, A. F., Grosvenor, J. A. H., Smith, K. R., Reblin,
529 J. S., Stutz, J., Richardson, A. D., Logan, B. A., & Bowling, D. R. (2024). The
530 biological basis for using optical signals to track evergreen needleleaf
photosynthesis. *BioScience*, 74(3), 130–145.
<https://doi.org/10.1093/biosci/biad116>

Rajagopalan, K., Chinnayakanahalli, K. J., Stockle, C. O., Nelson, R. L., Kruger, C. E.,
Brady, M. P., Malek, K., Dinesh, S. T., Barber, M. E., Hamlet, A. F., Yorgey, G.
G., & Adam, J. C. (2018). Impacts of Near-Term Climate Change on Irrigation
Demands and Crop Yields in the Columbia River Basin. *Water Resources*
*Research*, 54(3), Article 3. <https://doi.org/10.1002/2017WR020954>

Swain, D. L. (2021). A Shorter, Sharper Rainy Season Amplifies California Wildfire Risk.
*Geophysical Research Letters*, 48(5), e2021GL092843.
<https://doi.org/10.1029/2021GL092843>

Swain, D. L., Horton, D. E., Singh, D., & Diffenbaugh, N. S. (2016). Trends in
atmospheric patterns conducive to seasonal precipitation and temperature
extremes in California. *Science Advances*, 2(4), e1501344.
<https://doi.org/10.1126/sciadv.1501344>

Swain, D. L., Prein, A. F., Abatzoglou, J. T., Albano, C. M., Brunner, M., Diffenbaugh, N.
S., Singh, D., Skinner, C. B., & Touma, D. (2025). Hydroclimate volatility on a
warming Earth. *Nature Reviews Earth & Environment*, 6(1), 35–50.
<https://doi.org/10.1038/s43017-024-00624-z>

United States Department of Agriculture Natural Resources Conservation Service.
(2023). *CA Water Supply Outlook Report—May 2023*.

<https://www.nrcs.usda.gov/sites/default/files/2023-02/CA->
[Water%20Supply%20Outlook%20Report-Feb%202023.pdf](https://www.nrcs.usda.gov/sites/default/files/2023-02/CA-Water%20Supply%20Outlook%20Report-Feb%202023.pdf)

United States Geological Survey. (2023). *Annual National Land Cover Database*
*(NLCD) Collection 1 Products* [Pdf,png]. U.S. Geological Survey.
<https://doi.org/10.5066/P94UXNTS>

Volk, J. M., Huntington, J. L., Melton, F. S., Allen, R., Anderson, M., Fisher, J. B., Kilic,
557 A., Ruhoff, A., Senay, G. B., Minor, B., Morton, C., Ott, T., Johnson, L., Comini
de Andrade, B., Carrara, W., Doherty, C. T., Dunkerly, C., Friedrichs, M.,
Guzman, A., ... Yang, Y. (2024). Assessing the accuracy of OpenET satellite-
based evapotranspiration data to support water resource and land management
applications. *Nature Water*, 2(2), 193–205. [https://doi.org/10.1038/s44221-023-](https://doi.org/10.1038/s44221-023-00181-7)
[00181-7](https://doi.org/10.1038/s44221-023-00181-7)

Williams, A. P., Cook, E. R., Smerdon, J. E., Cook, B. I., Abatzoglou, J. T., Bolles, K.,
Baek, S. H., Badger, A. M., & Livneh, B. (2020). Large contribution from
anthropogenic warming to an emerging North American megadrought. *Science*
*(New York, N. Y.)*, 368(6488), 314–318. <https://doi.org/10.1126/science.aaz9600>

Xia, Y., Hobbins, M. T., Mu, Q., & Ek, M. B. (2015). Evaluation of NLDAS-2
evapotranspiration against tower flux site observations. *Hydrological Processes*,
29(7), Article 7. <https://doi.org/10.1002/hyp.10299>

Zhang, T., Lin, X., Rogers, D. H., & Lamm, F. R. (2015). Adaptation of Irrigation
Infrastructure on Irrigation Demands under Future Drought in the United States.
*Earth Interactions*, 19(7), Article 7. <https://doi.org/10.1175/EI-D-14-0035.1>

Zhao, M., A, G., Liu, Y., & Konings, A. G. (2022). Evapotranspiration frequently

increases during droughts. *Nature Climate Change*, 12(11), 1024–1030.
<https://doi.org/10.1038/s41558-022-01505-3>